# *Drosophila* Nanos acts as a molecular clamp that modulates the RNA-binding and repression activities of Pumilio

Chase A Weidmann[1†], Chen Qiu[2,3†], René M Arvola[1], Tzu-Fang Lou[4], Jordan Killingsworth[1], Zachary T Campbell[4], Traci M Tanaka Hall[2,3], Aaron C Goldstrohm[1,5*]

[1]Department of Biological Chemistry, University of Michigan, Ann Arbor, United States; [2]Epigenetics and Stem Cell Biology Laboratory, National Institutes of Health, Research Triangle Park, United States; [3]National Institute of Environmental Health Sciences, National Institutes of Health, Research Triangle Park, United States; [4]Department of Biological Sciences, University of Texas at Dallas, Richardson, United States; [5]Department of Biochemistry, Molecular Biology and Biophysics, University of Minnesota, Minneapolis, United States

*For correspondence: agoldstr@umn.edu

[†]These authors contributed equally to this work

Competing interests: The authors declare that no competing interests exist.

**Abstract** Collaboration among the multitude of RNA-binding proteins (RBPs) is ubiquitous, yet our understanding of these key regulatory complexes has been limited to single RBPs. We investigated combinatorial translational regulation by *Drosophila* Pumilio (Pum) and Nanos (Nos), which control development, fertility, and neuronal functions. Our results show how the specificity of one RBP (Pum) is modulated by cooperative RNA recognition with a second RBP (Nos) to synergistically repress mRNAs. Crystal structures of Nos-Pum-RNA complexes reveal that Nos embraces Pum and RNA, contributes sequence-specific contacts, and increases Pum RNA-binding affinity. Nos shifts the recognition sequence and promotes repression complex formation on mRNAs that are not stably bound by Pum alone, explaining the preponderance of sub-optimal Pum sites regulated *in vivo*. Our results illuminate the molecular mechanism of a regulatory switch controlling crucial gene expression programs, and provide a framework for understanding how the partnering of RBPs evokes changes in binding specificity that underlie regulatory network dynamics.

## Introduction

Post-transcriptional gene regulatory mechanisms are widespread and mediated by hundreds of RNA-binding proteins (RBPs) that interact dynamically with target mRNAs (*Baltz et al., 2012*; *Castello et al., 2012*; *Gerstberger et al., 2014*). Understanding how RBPs function together to control the location, timing and level of protein expression is paramount. Decades of research have established the crucial roles of two archetypal RBPs, *Drosophila melanogaster* Nanos (Nos) and Pumilio (Pum), in developmental patterning, fertility, and nervous system development and function. We use this system to explore how two RBPs cooperatively define regulatory networks.

Pum is a founding member of the Pum/*fem-3* mRNA-binding factor (FBF), or PUF family, of eukaryotic RBPs (*Wickens et al., 2002*). PUF proteins share a sequence-specific RNA-binding domain known as the Pum Homology Domain (PUM-HD) (*Barker et al., 1992*; *Macdonald, 1992*; *Wharton et al., 1998*; *Zamore et al., 1997*; *Zhang et al., 1997*). Crystal structures of PUF proteins have illuminated their unique RNA recognition properties (*Edwards et al., 2001*; *Miller et al., 2008*; *Qiu et al., 2012*; *Wang et al., 2001*, *2002*, *2009*; *Wilinski et al., 2015*; *Zhu et al., 2009*). Pum

**eLife digest** Molecules of DNA contain the instructions needed to make proteins inside cells. Proteins perform many different roles and each needs to be produced at the right time and in the right amounts to enable the cells to survive. The DNA is first copied to make molecules of ribonucleic acid (RNA), which are then used as templates to make the proteins. One way to control protein production is to regulate the RNA molecules. A family of proteins called RNA-binding proteins can recognise and bind to specific RNA molecules and determine whether a RNA molecule is destroyed, used to produce proteins, or stored for later use. In effect, these RNA-binding proteins act as switches that turn protein production on or off.

Nanos and Pumilio are two RNA-binding proteins that are found in many organisms, including humans and other animals. Genetic studies in fruit flies show that these two proteins influence development, the nervous system and the behaviour of stem cells by switching off the production of certain proteins. To investigate how Nanos and Pumilio work together to regulate protein production, Weidmann, Qiu et al. used a variety of techniques to study the activity of these proteins in cells taken from fruit fly embryos.

The experiments reveal that Nanos acts like a clamp to hold Pumilio close to specific RNAs, which allows Pumilio to switch off the production of the corresponding proteins. The presence of Nanos allows Pumilio to regulate RNAs that it cannot bind to alone. Therefore, the experiments show that by working together with Nanos, Pumilio is able to regulate a wider variety of RNAs than it would otherwise be able to. These findings provide a molecular understanding for why fruit fly mutants that lack Nanos or Pumilio have severe body defects and reduced fertility. The next challenge is to identify the specific RNAs targeted by Nanos and Pumilio in stem cells, the nervous system and during development.

binds with high affinity to specific sequences in target mRNAs and represses protein expression and enhances mRNA degradation (*Gerber et al., 2006*; *Weidmann et al., 2012*, *2014*; *Wharton et al., 1998*; *Wreden et al., 1997*; *Zamore et al., 1999*, *1997*). A classic example of Pum activity is the establishment of embryonic body pattern through repression of maternal *hunchback* (*hb*) mRNA, which requires collaboration with Nos (*Lehmann and Nusslein-Volhard, 1987*, *1991*; *Murata and Wharton, 1995*; *Sonoda and Wharton, 1999*; *Wang and Lehmann, 1991*).

Regulation of *hb* mRNA by Pum relies on the spatial distribution of Nos, a tandem CCHC Zn finger (ZF) RBP (*Barker et al., 1992*; *Curtis et al., 1997*; *Forbes and Lehmann, 1998*). Maternal *hb* mRNA and Pum protein are distributed throughout the syncytial *Drosophila* embryo (*Macdonald, 1992*; *Tautz, 1988*), whereas Nos protein forms a gradient emanating from the posterior end (*Wang and Lehmann, 1991*). Where their expression overlaps, Pum and Nos together repress *hb* mRNA (*Barker et al., 1992*; *Lehmann and Nusslein-Volhard, 1991*; *Murata and Wharton, 1995*; *Sonoda and Wharton, 1999*). In the absence of Nos or Pum expression, Hb protein is produced throughout the embryo, and no abdominal segments are formed.

Regulation of *hb* mRNA by Pum and Nos is dependent on two Nanos Response Elements (NREs) in the 3′UTR (*Murata and Wharton, 1995*; *Wharton and Struhl, 1991*). Each NRE contains a binding site for Pum with the RNA consensus sequence, 5′-UGUAHAUA (where H is A, U or C), the Pumilio Response Element (PRE). Additional nucleotides in the NRE, located 5′ of the PRE, are functionally important for *hb* regulation and were proposed to be recognized by Nos (*Edwards et al., 2001*; *Sonoda and Wharton, 1999*). By itself, Nos appeared to lack RNA sequence specificity (*Curtis et al., 1997*). Instead, Nos binding to *hb* NREs requires Pum RNA recognition (*Sonoda and Wharton, 1999*). Hence, combinatorial *hb* mRNA repression requires the sequence specificity of Pum and the spatial information provided by the Nos gradient.

Nos and Pum also regulate germline and neurological processes (*Forbes and Lehmann, 1998*; *Mee et al., 2004*; *Menon et al., 2009*, *2004*; *Ye et al., 2004*). Nos and Pum collaborate to repress expression of *Cyclin B* mRNA (*CycB*) in primordial germ cells and germline stem cells (*Asaoka-Taguchi et al., 1999*) and the sodium channel *paralytic* (*para*) in the nervous system (*Muraro et al., 2008*). Like *hb* mRNA, *CycB* and *para* mRNAs possess NREs with PRE-like motifs. Furthermore,

genome-wide analyses have identified hundreds of Pum-associated mRNAs, suggesting that Pum may play an expansive regulatory role beyond the few validated target RNAs (*Gerber et al., 2006*; *Laver et al., 2015*).

While collaboration between Nos and Pum is firmly established, the mechanism by which they do so remains to be determined. Here, we report the crystal structures of Nos-Pum-RNA complexes, which reveal that Nos acts as a molecular clamp that embraces both Pum and RNA. The C-terminal region of Pum undergoes conformational changes to make new contacts with the RNA and Nos. We explored the hypothesis that Nos promotes repression by modulating the RNA-binding activity of Pum. We show that Nos enhances the cellular repression activity and in vitro RNA-binding affinity of Pum. Moreover, Nos contacts nucleotides upstream of the PRE. In doing so, Nos alters the specificity of the repression complex and promotes repression of RNAs that are not stably bound by Pum alone. We performed RNA target selection and high-throughput sequencing, which, together with RNA-binding and cellular repression assays, demonstrate that Nos diversifies Pum RNA regulatory networks.

## Results

### Nanos Zn Finger and C-terminal regions collaborate with Pumilio to repress target gene expression

We established a cell-based *hb* reporter mRNA assay, wherein exogenous Nos robustly repressed reporter protein and RNA expression in a manner dependent on the PREs (*Figure 1*) and Pum (*Weidmann and Goldstrohm, 2012*). We used D.mel-2 cells, which do not express Nos and express insufficient Pum to repress the reporter efficiently in the absence of exogenous Nos (*Weidmann and Goldstrohm, 2012*), and a Renilla luciferase (RnLuc) reporter containing the 3′UTR of *hb* with two NREs, each of which possesses a PRE (*Figure 1A*). Halo-tagged, full-length Nos protein, comprising N-terminal, tandem ZF (Z), and C-terminal regions (NZC, *Figure 1B*), repressed reporter expression 75% relative to a negative control Halo-tag (Halo) protein alone (*Figure 1C*). Mutation of one PRE modestly reduced repression (*Figure 1C*). In contrast, mutation of both PREs abrogated repression. Nos also reduced the level of reporter mRNA (*Figure 1—figure supplement 1*), consistent with enhanced degradation of target mRNAs. Together, these data show that Nos-enhanced, Pum-mediated repression is PRE-dependent, and a single NRE is sufficient to confer regulation.

Using this approach, we performed structure-function analysis of Nos to identify which regions are required for activity. Conserved tandem ZFs are necessary for RNA binding (*Curtis et al., 1997*; *Sonoda and Wharton, 1999*), and mutations that disrupt either ZF (C319Y or C354Y) completely blocked Nos repression activity (*Figure 1D*). Similarly, a Nos variant lacking the C-terminal region (NZ) displayed limited repression activity, 16% repression versus 79% for full-length Nos (NZC). In contrast, a Nos variant lacking the N-terminal region (ZC) retained 43% repression. No individual Nos region (N, Z, or C) was sufficient for repression. These findings indicate that multiple regions of Nos contribute to enhancing Pum-mediated repression: the ZFs and the C-terminal region are crucial, and the N-terminal region makes a minor contribution. Because the Nos ZC region retained most of the Nos-enhanced repression activity and was amenable to biochemical and structural studies, we focused on determining its mechanism of action.

### Structural basis for interactions between Nanos, Pumilio and RNA

We determined a 3.7 Å crystal structure that reveals the molecular architecture of the Nos-Pum-RNA ternary complex (*Table 1*, *Figure 2A*, and *Figure 2—figure supplement 1*). Strikingly, the structure illustrates how the Nos tandem ZFs envelop the RNA bases immediately upstream of the PRE and, together with the C-terminal region, embrace both the RNA and Pum (*Figure 2A and B*). We crystalized Pum and Nos with a *hb* NRE2 RNA, 5′-AAAU<u>UGUACAUA</u>AGCC (the core PRE sequence is underlined, and we designate the first U of the core PRE as position +1 and number the four upstream nucleotides as −1 to −4). The ternary complex structure reveals critical protein-protein and protein-RNA interactions between Nos, Pum and RNA. We also determined a 1.14 Å crystal structure of a binary complex of Pum and a *hb* PRE2 RNA (5′-UGUACAUA) (*Figure 2C*), which exemplifies the modular 1 repeat:1 RNA base recognition of classical PUF proteins. For example, the UGU motif of the PRE is recognized by repeats R8 through R6. This high-resolution structure,

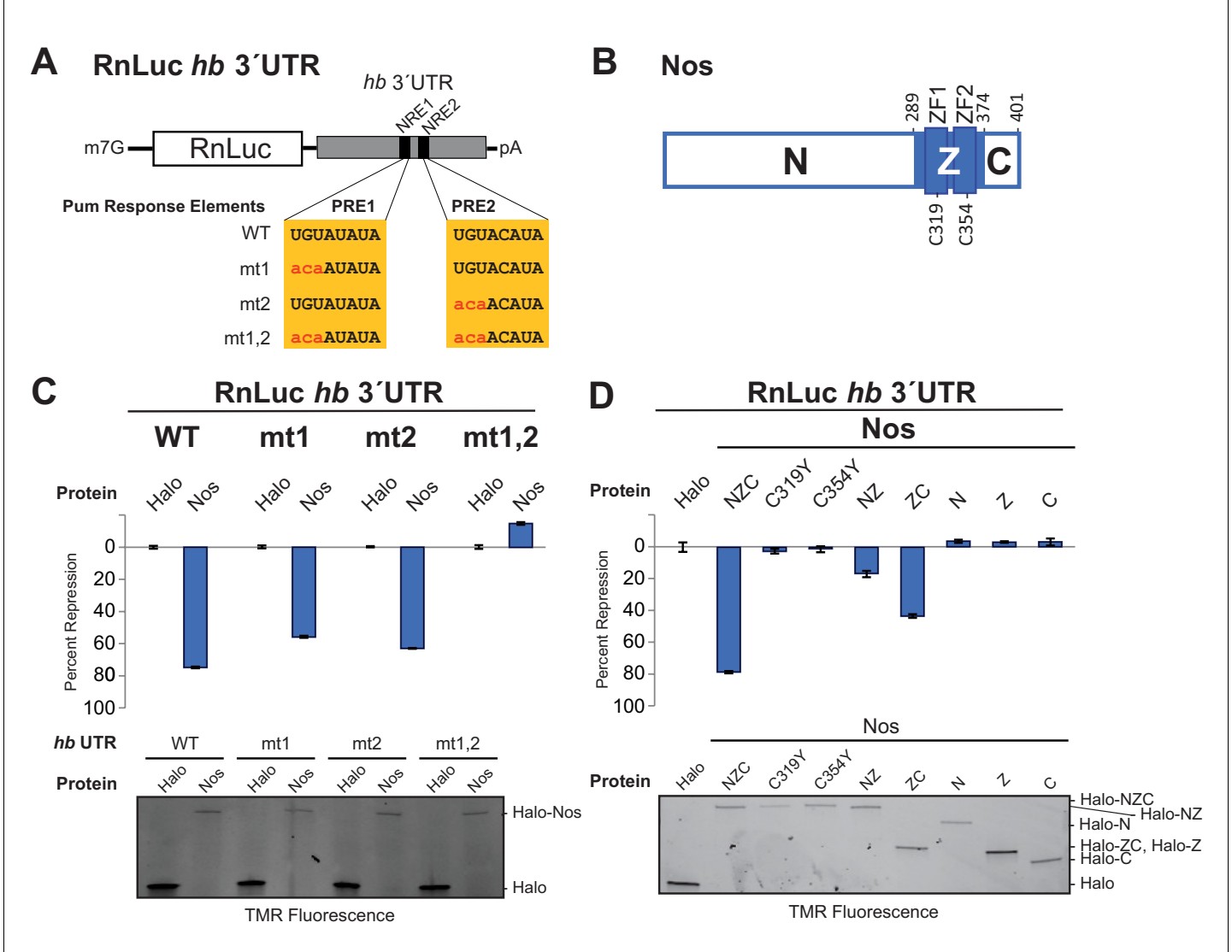

**Figure 1.** The Zn finger and C-terminal regions of Nanos collaborate with Pumilio to repress target protein and mRNA expression. (A) Diagram of Renilla Luciferase (RnLuc) reporters including the *hb* 3′UTR. WT PRE1 and PRE2 sequences, located within NRE1 and NRE2, respectively, and mutant PRE sequences (mt1, mt2, and mt1,2) are shown. (B) Diagram of Nos protein. Amino acid residue boundaries of the N-terminal region (N), central region (Z, blue) including ZFs, and C-terminal extension (C) are indicated. (C) Nos-enhanced repression via the *hb* PREs. Reporter assays were performed in D.mel-2 cells. Percent repression values are graphed for RnLuc WT and mt *hb* 3′UTR reporter expression with negative control Halo-tag alone (Halo) and full length Halo-Nos test proteins. (D) The Nos Z and C-terminal regions together retain efficient repression activity. Percent repression values are graphed for RnLuc *hb* 3′UTR WT reporter expression with negative control Halo and Halo-Nos variants are shown. For panels **C** and **D**, mean and standard error of the mean (SEM) values from quadruplicate experiments are shown. Expression of test proteins was visualized by tetramethylrhodamine (TMR) fluorescent labeling of the Halo-tag fusion proteins. Statistical analysis of the data is reported in *Figure 1—source data 1*.

The following source data and figure supplements are available for figure 1:

**Source data 1.** Values and statistical analysis of luciferase reporter assays.
**Figure supplement 1.** Nos reduces hb 3′UTR reporter mRNA level in a PRE-dependent manner.
**Figure supplement 1—source data 1.** Values and statistical analysis of Northern blot of luciferase reporter mRNAs.

**Table 1.** Data collection and refinement statistics.

| | | Pum-RNA | Pum-Nos-*hb* RNA | Pum-Nos-*cycB* RNA |
|---|---|---|---|---|
| PDB ID | | 5KLA | 5KL1 | 5KL8 |
| Data collection | | | | |
| Space group | | C2 | P6$_5$22 | P6$_5$22 |
| Cell dimensions | *a, b, c* (Å) | 194.9, 29.5, 62.0 | 137.0, 137.0, 221.4 | 135.1, 135.1, 220.4 |
| | a, b, g (°) | 90.0, 101.2, 90.0 | 90.0, 90.0, 120.0 | 90.0, 90.0, 120.0 |
| Resolution (Å) | | 50-1.14 (1.16-1.14) | 50-3.70 (3.83-3.70) | 50-4.00 (4.12-4.00) |
| $R_{sym}$ | | 0.045 (0.387) | 0.128 (0.747) | 0.143 (0.779) |
| $I / \sigma I$ | | 36.9 (2.7) | 19.1 (2.8) | 13.0 (3.6) |
| Completeness (%) | | 99.7 (97.4) | 99.3 (93.2) | 99.3 (100.0) |
| Redundancy | | 4.2 (2.4) | 11.3 (11.0) | 8.9 (8.7) |
| Refinement | | | | |
| Resolution (Å) | | 34.46 - 1.14 | 38.3 - 3.70 | 39.0 - 4.00 |
| No. reflections | | 127077 | 13562 | 10715 |
| $R_{work}$ / $R_{free}$ (%) | | 16.0 / 17.4 | 26.4 / 30.0 | 28.3 / 31.2 |
| No. atoms | | | | |
| Protein | | 5532 | 3194 | 3021 |
| RNA | | 253 | 252 | 226 |
| Water / Solvent | | 401 | 0 | 0 |
| *B*-factors | | | | |
| Protein | | 29.0 | 175.5 | 208.6 |
| RNA | | 20.5 | 150.4 | 183.4 |
| Water / Solvent | | 34.9 | - | - |
| R.m.s deviations | | | | |
| Bond lengths (Å) | | 0.007 | 0.003 | 0.002 |
| Bond angles (°) | | 0.950 | 0.605 | 0.508 |

*Values in parentheses are for highest-resolution shell.

combined with previous crystal structures of *Drosophila* Pum (*Edwards et al., 2001*) and zebrafish Nos ZFs (*Hashimoto et al., 2010*), allowed us to build and refine the ternary complex model.

Comparison of the ternary and binary complexes reveals that the addition of Nos and the upstream nucleotides induces localized conformational changes in Pum that promote Nos-Pum interaction and binding of Pum to RNA upstream of the core PRE site. While the overall structure of Pum in the ternary complex is similar to that in the binary complex (RMSD of 1.2 Å over 324 Pum Cα atoms), the C-terminal region of Pum undergoes notable changes. Loop residues between repeats R7 and R8 rearrange in the ternary complex to promote interaction of F1367$_{Pum}$ with the C-terminal α helix of Nos (*Figure 2—figure supplement 2*). In addition, the C-terminal α helix of Pum (helix α2 of repeat R8′) unfolds to promote interaction of residues with the upstream RNA backbone (*Figure 2—figure supplement 2* and *Figure 2—figure supplement 3*). Since the upstream nucleotides were not present in the binary complex, the change in the structure of the C-terminal region of Pum may be induced by the presence of the upstream RNA and/or Nos. With these conformational changes, Pum and Nos interact with one another and together recognize RNA immediately 5′ of the core PRE motif.

## Nanos increases the binding affinity of Pumilio for *hunchback* RNA

Using an electrophoretic mobility shift assay (EMSA), we demonstrated that Nos ZC binds stably and tightly to a Pum-*hb* NRE2 RNA complex and cooperatively strengthens the binding affinity of Pum

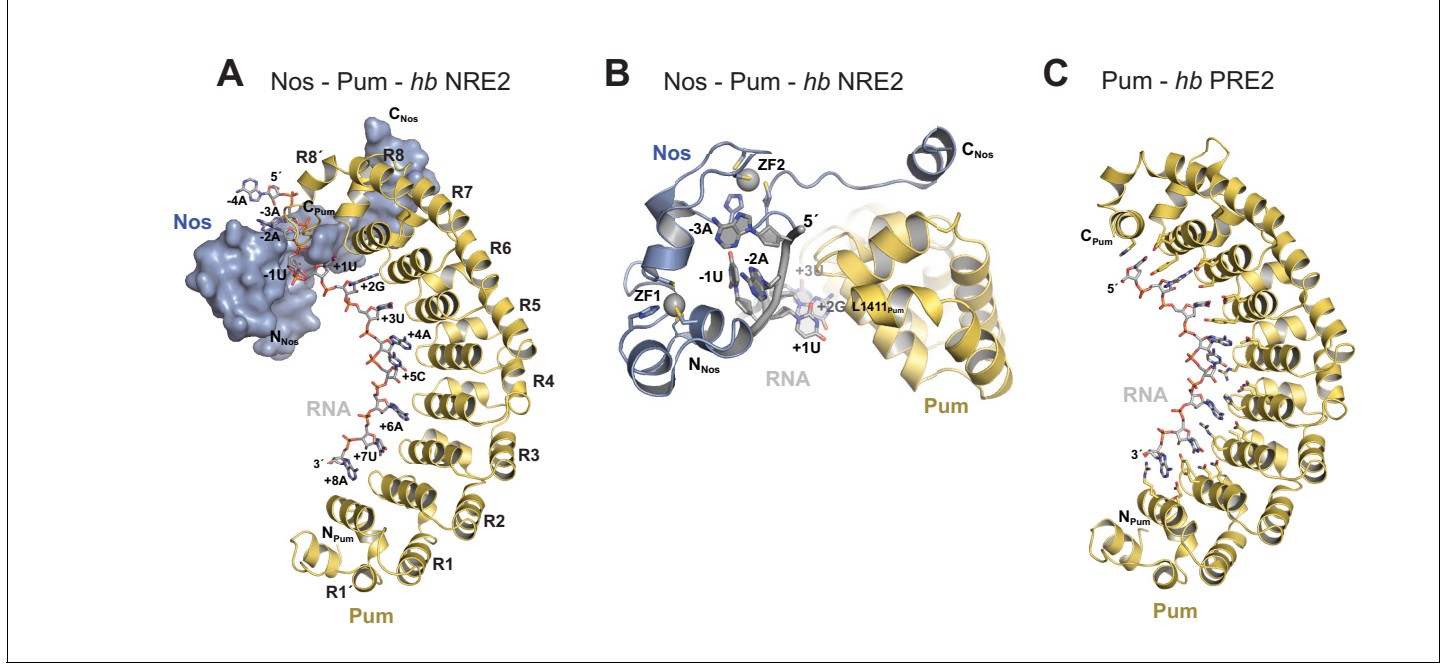

**Figure 2.** Nanos embraces Pumilio and *hb* RNA to stabilize the ternary complex. (**A**) Crystal structure of a *Drosophila* Nos-Pum-*hb* NRE2 RNA ternary complex. Pum is shown as a ribbon diagram (yellow), Nos is shown as a surface representation (blue) and *hb* NRE2 RNA is shown as a stick model colored by atom type. N- and C-termini of proteins and 5´ and 3´ ends of the RNA are labeled. Pum repeats R1 to R8 and pseudo repeats R1´ and R8´ are indicated. (**B**) View of the Nos-Pum-*hb* NRE2 RNA ternary complex down the long axis of Pum. Pum (yellow) and Nos (blue) are shown as ribbon diagrams, and *hb* NRE2 is shown as a cartoon backbone with RNA bases. Zn atoms are shown as grey spheres. The C-terminus of Pum was truncated to L1411 to allow visualization of Nos-RNA interaction. (**C**) Ribbon diagram of a crystal structure of a *Drosophila* Pum binary complex with the PRE2 from *hb* NRE2.

The following figure supplements are available for figure 2:

**Figure supplement 1.** Representative electron density map of the Nos-Pum-*hb* NRE2 complex.

**Figure supplement 2.** Nanos induces localized structural changes in Pum upon formation of the Nos-Pum-*hb* NRE2 RNA ternary complex.

**Figure supplement 3.** Crystal structure of Nos – Pum – hb NRE2 RNA ternary complex highlights key Pum-RNA and Nos-Pum contacts.

for *hb* RNA. The RNA-binding domain of Pum (PUM-HD) bound *hb* NRE2 RNA, and addition of equimolar Nos ZC to the binding reaction further retarded the *hb* NRE2 RNA mobility, indicating formation of a Nos-Pum-RNA ternary complex (*Figure 3A,B*, and *Figure 3—figure supplement 1*). Disruption of either ZF of Nos (C319Y or C354Y) eliminated ternary complex formation (*Figure 3B*). In addition, the Pum-*hb* NRE2 RNA interaction is essential for Nos binding, as RNA-binding deficient Pum mutR7 did not support a ternary complex (*Figure 3B*). Nos did not shift *hb* NRE2 RNA on its own (*Figure 3B*), even at protein concentrations of one micromolar (*Figure 3C*). We applied the EMSA quantitatively and found that Nos binds with high affinity to the Pum-*hb* NRE2 RNA complex (*Figure 3C and D*) and increases Pum binding affinity for *hb* NRE2 RNA by 3-fold (*Figure 3E and F*). These data establish that the requirements for cooperative assembly of the Nos-Pum-*hb* NRE2 complex in vitro mirror those for Nos-enhanced, Pum-mediated repression in cells and embryos, and therefore complex formation reflects repression activity.

## Interactions between Nanos and Pumilio are necessary for repression

The protein-protein interactions observed between Nos and Pum are focused between the C-terminal end of Nos (I376$_{Nos}$ to E385$_{Nos}$) and the non-RNA-binding convex surface of Pum in repeats R7 and R8 (*Figure 4A and B*). For example, the side chain of Q1337$_{Pum}$ is within hydrogen bonding

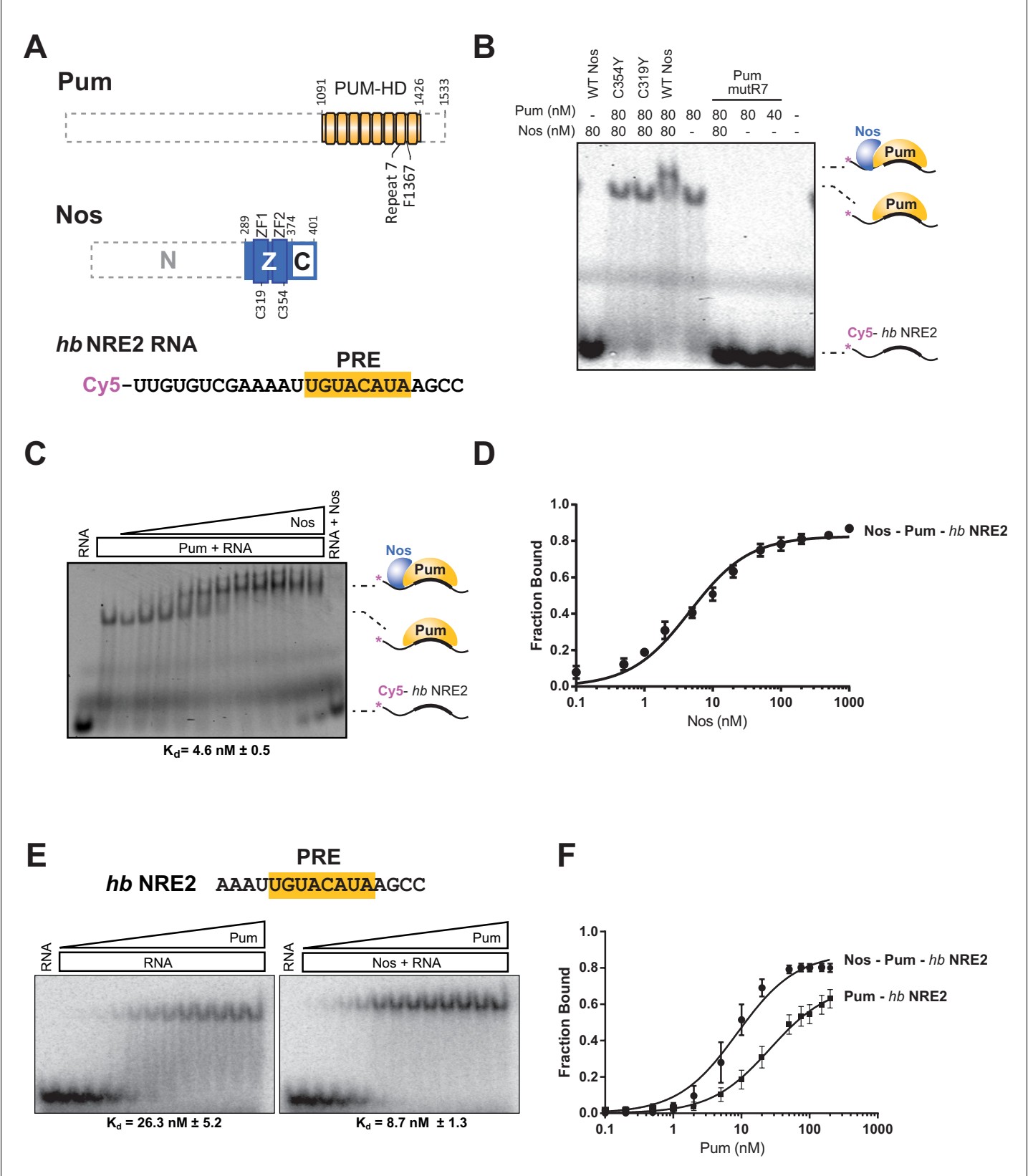

**Figure 3.** Nanos increases the binding affinity of Pumilio for *hunchback* RNA. (**A**) Diagram of recombinant proteins and RNA ligand used for EMSAs. The amino acid residue boundaries of the Pum RNA-binding domain (PUM-HD, yellow) are represented relative to full-length Pum. For simplicity, we

*Figure 3 continued*

refer to the PUM-HD as Pum. The Z and C regions are shown in the context of full-length Nos. Dashed lines outline regions excluded from the recombinant proteins used for EMSAs. The RNA sequence of the Cy5-labeled *hb* NRE2 is shown with the PRE sequence highlighted (yellow). (**B**) A representative EMSA with *hb* NRE2 RNA is shown. Nos and Pum test protein concentrations are indicated above the gel. (**C**) A representative EMSA measuring binding of Nos to the Pum – *hb* NRE2 complex. Nos was titrated into binding reactions with a constant concentration of Pum (100 nM). The mean observed dissociation constant ($K_d$) with standard deviation (SD) from triplicate experiments is shown below the gel. (**D**) Graph of fraction bound for Nos-Pum-*hb* NRE2 complex in response to Nos concentration. Mean and standard error of the mean (SEM) values from triplicate EMSAs are plotted. (**E**) Representative EMSAs measuring binding to *hb* NRE2 RNA, performed at the same time under identical conditions, titrating Pum in the presence or absence of Nos. The mean $K_d$ with SD from triplicate experiments is shown below the gel. (**F**) Graph of fraction bound of Nos-Pum-*hb* NRE2 and Pum – *hb* NRE2 complexes in response to Pum concentration. Mean and SEM values from triplicate EMSAs are plotted.

The following figure supplement is available for figure 3:

**Figure supplement 1.** Recombinant purified Pum and Nos test proteins.

distance of main chain N and O atoms of I376$_{Nos}$. In addition, F1367$_{Pum}$ forms part of a hydrophobic pocket that interacts with M378$_{Nos}$ in the Nos C-terminus(*Figure 4B*).

To examine the roles of the interaction between Nos and Pum for repression activity and complex formation, we measured the effects of targeted deletions of Nos on repression activity and found that interactions between the Nos C-terminal α helix and Pum are critical for repression. Since deletion of the C-terminal region severely limited Nos repression of the *hb* 3′UTR reporter in cells (*Figure 1D*), we probed this interaction more precisely. We tested a Δ376–382 deletion, which eliminated much of the Pum-binding interface and corresponds to the lesion in the *nos*$^{L7}$ allele that disrupts Nos function *in vivo* (*Curtis et al., 1997*). This deletion impaired Nos repression activity to a similar extent as deletion of the entire C-terminal region: 27% repression compared to 14% repression for Nos NZ (*Figure 4C*). Since our crystal structure indicates that the deleted region includes part of the C-terminal helix that interacts with Pum, it is possible that the protein, although expressed, could be incorrectly folded. We further examined this region by introducing single amino acid substitutions, including the I376$_{Nos}$ and M378$_{Nos}$ residues that contact Pum in the structure (*Figure 4B*). Repression activity of Nos I376A was diminished to 46%, and more so for Nos M378A (29%), whereas Nos I382A caused a modest decrease to 60%, relative to 70% repression by wild type Nos (*Figure 4D*). A Δ383–393 deletion, which removed the final three ordered residues in the structure and eight subsequent residues, also diminished repression activity to 43% (*Figure 4C*). In contrast, a Δ394–401 deletion retained full repression activity (*Figure 4C*). These C-terminal eight residues of Nos were disordered in the ternary complex structure, and therefore did not contact Pum. These results confirm the functional importance of the observed protein contacts between Pum and Nos for regulation in cells.

We also found that single amino acid substitutions in Pum disrupt formation of the repression complex. No ternary complex was formed with Pum F1367S (*Figure 4E*), an R7-R8 loop mutant that binds RNA, but does not interact with Nos in a yeast 3-hybrid assay (*Edwards et al., 2001*) or respond to Nos in cells (*Weidmann and Goldstrohm, 2012*). Similarly, a Q1337A$_{Pum}$ mutation eliminated ternary complex formation (*Figure 4F*). Importantly, both Pum mutants retained the ability to bind to *hb* NRE2 RNA (*Figure 4E and F*). Thus, Nos must interact with both repeat R7 and the R7-R8 loop of Pum to form a stable ternary complex.

## Interaction of Nanos Zn fingers with RNA extends the binding site and is critical for repression

The crystal structure of the Nos-Pum-*hb* RNA complex reveals that Nos binds to three nucleotides upstream of the core PRE when it joins the ternary complex, and we find that repression activity is highly sensitive to mutation of the interface. The first base upstream of the PRE, -1U, is inserted into a hydrophobic binding pocket formed by F321$_{Nos}$, T366$_{Nos}$, and Y369$_{Nos}$ (*Figure 5A* and *Figure 2—figure supplement 1*). The O4 atom of −1U is near the main chain N atom of T366$_{Nos}$. Nos also contacts the bases and backbone atoms of −2A and −3A (*Figure 5B*). Three residues within the rearranged C-terminal region of Pum, T1415$_{Pum}$, K1377$_{Pum}$, and K1413$_{Pum}$, appear to approach the phosphate groups of −2A, −3A, and −4A, respectively (*Figure 2—figure supplement*

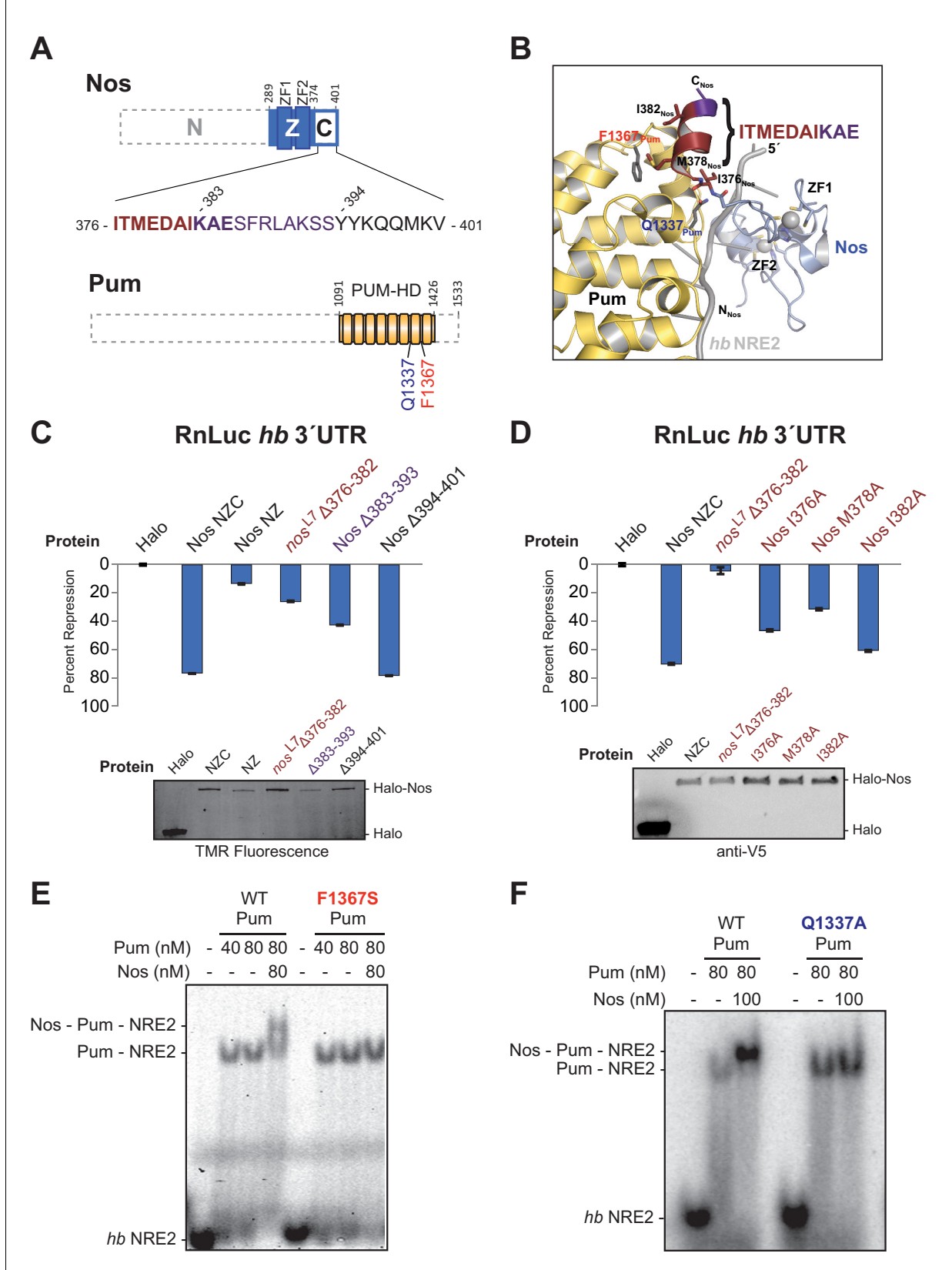

**Figure 4.** Interactions between Nanos and Pumilio are necessary for repression. (**A**) Diagrams of Nos and Pum proteins highlighting residues involved in protein-protein interaction. The amino acid sequence of the C-terminal region of Nos is shown. Residues 376–382 that are deleted in the *nos^L7* fly

*Figure 4 continued on next page*

Figure 4 continued

mutant, a strong allele for defective abdominal segmentation, are colored red, and residues 383–393 are colored purple. Residues that form the Nos C-terminal α helix are in boldface. (B) View of the interface between Nos (blue with red and purple C-terminal region) and Pum (yellow). Interacting residues in Nos and Pum are shown in stick representation, and the *hb* NRE2 RNA is shown as a cartoon representation. (C and D) Percent repression values are graphed for the RnLuc WT *hb* 3′UTR reporter with negative control Halo-tag alone (Halo) and variants of Halo-Nos test proteins. Nos test proteins included full-length Nos (NZC), a truncation of the C-terminal region (NZ), deletions (Δ) of the indicated amino acids and specific amino acid substitutions in the context of full-length Nos. Labels are colored as in panels A and B. Mean and SEM values from quadruplicate samples are shown. Expression of test proteins was visualized by TMR fluorescent labeling or anti-V5 western blotting of the Halo-tag fusion proteins. Statistical analysis of the data is reported in *Figure 4—source data 1*. (E and F) Representative EMSAs comparing ternary complex formation by WT Pum or the mutant F1367S Pum (panel D) or the mutant Q1337A Pum (panel E).

The following source data is available for figure 4:

**Source data 1.** Values and statistical analysis of luciferase reporter assays.

*1* and *Figure 2—figure supplement 3*). Interaction of Nos and Pum with the RNA nucleotides upstream of the PRE explains how Nos strengthens the overall ternary complex.

To determine the importance of Nos-RNA interactions, we measured the effect of single amino acid substitutions on cellular repression activity (*Figure 5C*). Individual mutations that disrupted the hydrophobic binding pocket for the −1U base (F321A, T366A, and Y369A) abrogated repression activity. Another mutant, K368Q, designed to eliminate a salt bridge interaction with the phosphate group of −1U, reduced repression activity to 32% vs 75% for WT protein. Mutations targeting interactions with nucleotides−2A and −3A (N325A and Y352A) had smaller, but measurable effects on repression activity. The effects of these single residue substitutions indicate the importance of Nos recognition of the −1 nucleotide and interactions with other upstream NRE nucleotides.

We next investigated whether the sequence of the upstream nucleotides (defined structurally as the Nos binding site, NBS) is important for ternary complex formation and repression activity. Substitution of both the −1 and −2 positions of *hb* NRE2 with cytosine had been shown previously to disrupt abdominal segmentation, but did not affect Pum RNA association (*Murata and Wharton, 1995*), so we designed *hb* NRE2 RNAs that substituted cytosine bases at either the −1 or −2 position (*Figure 5D*, −1C and −2C). Neither mutation hindered RNA binding by Pum, but both mutations blocked ternary complex formation (*Figure 5E*). We then probed whether the cytosine substitutions in the NBS affect mRNA regulation in cells using reporters bearing a 3′UTR with a single *hb* NRE (RnLuc 1x *hb* NRE2). Expression of full-length Nos resulted in 50% repression of WT reporter activity compared to the negative control Halo-tag protein alone (*Figure 5F*), similar to a mutant *hb* 3′UTR reporter with only a functional NRE2 sequence (*Figure 1C*, mt1). In contrast, Nos did not repress mutant −1C or −2C reporters (*Figure 5F*), consistent with disruption of ternary complex formation by these substitutions. Thus, the identities of the nucleotides in the NBS are critical for repression activity. Although we did not observe sequence-specific contacts to the −2A base in our crystal structure of the ternary complex, this likely reflects the modest 3.7 Å resolution that was not sufficient to resolve all direct contacts or identify water molecules that may mediate protein-RNA interaction.

## Nanos alters Pumilio RNA-binding specificity and affinity

Given the importance of the NBS sequence for Nos-enhanced Pum regulation, we examined whether differences in natural Nos-Pum mRNA target sequences affect regulatory activity. Using EMSAs, we found that Nos induced formation of a Nos-Pum-RNA ternary complex, even when Pum alone did not bind stably to RNA. The *Cyclin B (CycB)* NRE contains a PRE that diverges from the consensus with uracils in place of adenines at positions +6 and +8, and it has a different NBS sequence (*Figure 6A*). Pum alone did not form a stable complex with *CycB* NRE RNA (*Figure 6A*, left, and *6B*), but remarkably, addition of Nos ZC resulted in ternary complex formation with an apparent $K_d$ of 12 nM (*Figure 6A*, right, and *6B*), similar to the 8.7 nM $K_d$ for *hb* NRE2. We next tested complex formation with sequences from the *hb* NRE1 (*Figure 6—figure supplement 1*) and *bicoid* NRE (*Figure 6—figure supplement 2*), which match the PRE consensus sequence, but have a uracil or guanine, respectively, at the +5 position. For each of these sequences, Pum unexpectedly

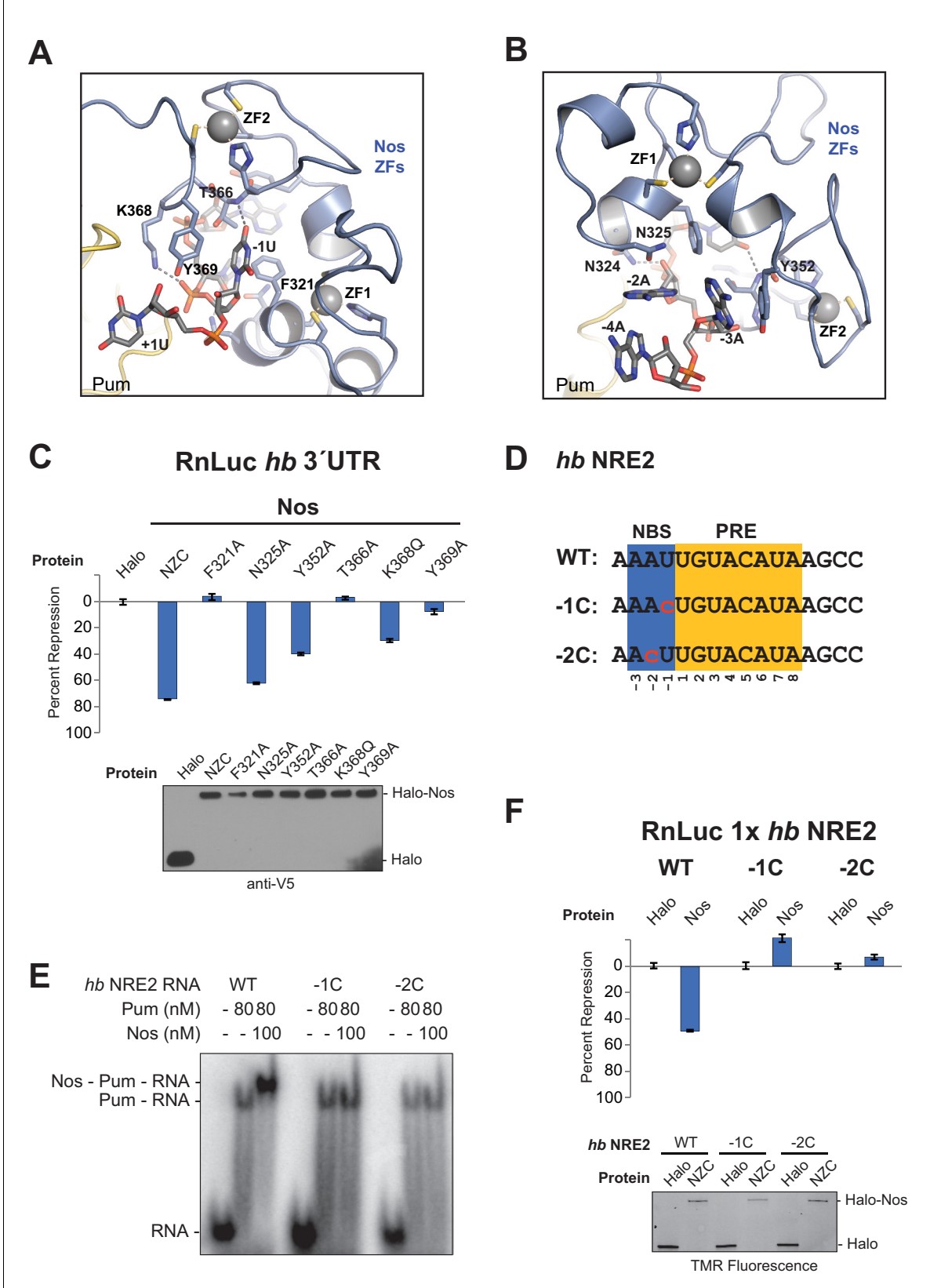

**Figure 5.** Nanos Zn finger interaction with RNA extends the RNA-binding site and is critical for repression. (**A**) Interaction of Nos ZFs with the -1U of *hb* NRE2. In addition to the base contacts noted in the text, the OH group of Y369$_{Nos}$ and the NH$_2$ group of K368$_{Nos}$ interact with the phosphate group of

*Figure 5 continued on next page*

*Figure 5 continued*

−1U. (B) Interaction of Nos ZFs with the −2A and −3A nucleotides. In panels A and B, important interactions between nucleotide and amino acid residues are shown. Zn atoms are shown as grey spheres with coordination by CCHC residues indicated by yellow dashed lines. (C) Percent repression values are graphed for the RnLuc WT *hb* 3′UTR reporter expression with negative control Halo-tag alone (Halo) and Halo-tag fusions of WT Nos or mutant Nos. Mutated residues are shown in panels A and B. Protein expression was confirmed by western blotting for the V5 epitope tag on each test protein. Statistical analysis of the data is reported in *Figure 5—source data 1*. (D) Sequences of *hb* NRE2 derivatives tested in EMSA (panel E) and reporter expression assay (panel F). The PRE core and the Nos binding site (NBS), derived from the crystal structure, are colored yellow and blue, respectively. Nucleotide changes in each RNA relative to the WT *hb* NRE2 sequence are marked by red lowercase letters. (E) Representative EMSA measuring ternary complex formation using indicated combinations of Nos and Pum with the RNA ligands shown in panel D. (F) Percent repression values for RnLuc reporters bearing a minimal 3′UTR containing a single *hb* NRE2 element or its mutant variants (panel D) with Halo or Halo-Nos are shown. Expression of test proteins was visualized by TMR fluorescent labeling of the Halo-tag fusion proteins. For panels C and F, mean and SEM values from quadruplicate samples are shown.

The following source data is available for figure 5:

**Source data 1.** Values and statistical analysis of luciferase reporter assays.

did not form a stable binary complex, but as with the *CycB* sequence, Nos induced ternary complex formation.

We then investigated the effect of specific RNA mutations on formation of the Nos-Pum-RNA ternary complex, focusing on the conserved UGU motif that is the hallmark of PUF protein binding sites. As expected, mutating +3U to G (*hb* PRE2 +3G) prevented stable binding of Pum alone (*Figure 6—figure supplement 3*). Yet, similar to the *Cyclin B* and *bicoid* NREs, addition of Nos ZC conferred formation of a ternary complex with a $K_d$ of 59.9 nM. In contrast, mutating UGU to ACA (*hb* PRE2 ACA) abolished both Pum binding and ternary complex formation (*Figure 6—figure supplement 4*). Taken together, our results demonstrate that Nos can stabilize binding of Pum to RNAs containing a wider range of consensus or divergent NREs, but cannot overcome complete disruption of the UGU trinucleotide sequence.

To gain molecular insight into how Nos enhances Pum recognition of different NRE sequences, we determined a 4.0 Å crystal structure of a ternary complex with a *CycB* NRE RNA (5′-UAUU<u>U-GUAAUUU</u>AU, core PRE underlined)(*Table 1*). The protein structures are essentially unchanged compared to the complex with *hb* NRE2 RNA; however, differences in Pum binding to bases +5 to +8 result in distinct conformations of *hb* and *CycB* RNAs in this region (*Figure 6C* and *Figure 6—figure supplement 5*). Pum binds to the *hb* PRE2 using the base-omission mode observed for human Pumilio1 (*Lu and Hall, 2011*), where bases +4 and +5 stack directly (*Figure 6D*). In contrast, Pum appears to bind to the *CycB* PRE using the 1 repeat:1 RNA base PUF recognition mode with R1271$_{Pum}$ sandwiched between bases +4 and +5 (*Figure 6E*). Pum binds specifically to the 3′ AUA sequence at positions +6 to +8 of *hb* PRE2 RNA (*Figure 6D*). However, for the *CycB* RNA, recognition of +6U is suboptimal and the +8U nucleotide is disordered in the ternary complex (*Figure 6E*). Thus, Nos stabilizes binding of Pum to RNAs that do not match the PRE consensus sequence in the 3′ half, reducing Pum specificity to allow regulation of a broader range of mRNA targets than Pum alone.

To define changes in Pum specificity induced by Nos, we examined sequence preferences using SEQRS (in vitro <u>se</u>lection, high-throughput <u>s</u>equencing of <u>R</u>NA, and <u>s</u>equence specificity landscapes) (*Campbell et al., 2012a*) (*Figure 7A*). Pum alone reproducibly enriched a motif matching the PRE consensus (*Figure 7B*, *Figure 7—figure supplement 1*). Strikingly, addition of Nos to immobilized Pum enriched A/U-rich sequences upstream of the 5′-UGUA of the PRE (*Figure 7C*, *Figure 7—figure supplement 2*, *3*), consistent with Nos recognition of the NBS. Sequence selection at the 3′ end of the Pum motif weakened. Comparison of control SEQRS analyses demonstrated specificity of the interactions. The Pum motif was highly enriched by wild-type Pum alone and, to a lesser degree, in the presence of Nos (*Figure 7D*), the Nos-Pum motif was highly enriched by the wild-type ternary complex, whereas Nos alone or the RNA-binding defective Pum did not enrich either motif.

We examined whether the Nos-Pum motif was enriched in Pum target mRNAs identified from *Drosophila* embryos and adults (*Gerber et al., 2006*) and observed enrichment of the Nos-Pum motif and, to a lesser degree, the Pum motif, in targets from both embryos and adults (*Figure 7E*,

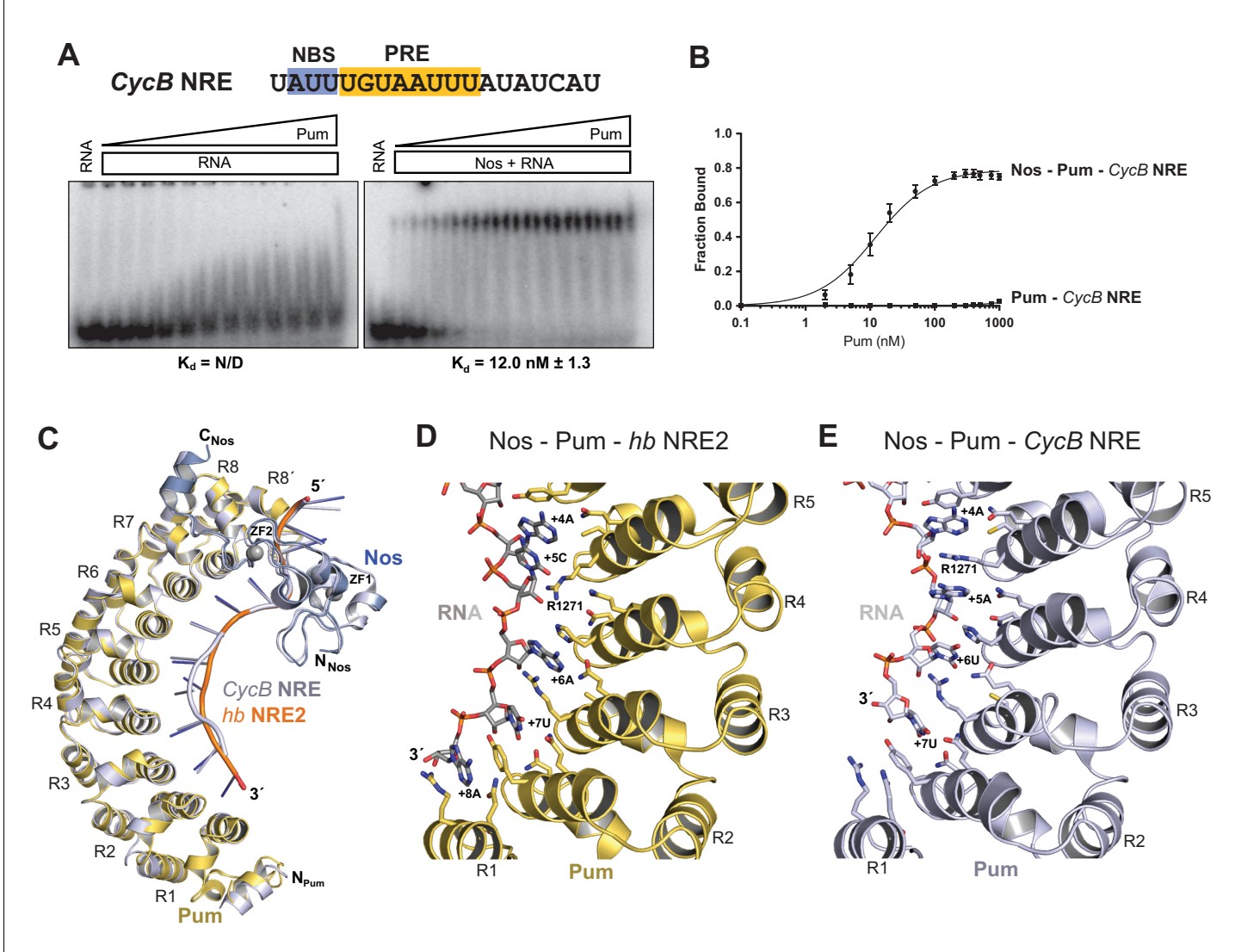

**Figure 6.** Nanos alters Pumilio RNA-binding specificity. (**A**) A representative EMSA with increasing concentrations of Pum in the presence or absence of Nos, performed with radiolabeled *CycB* NRE RNA shown at the top. The PRE and the NBS are highlighted in yellow and blue, respectively. The mean observed $K_d$ values with SD from triplicate experiments are shown below the gel. (**B**) Graph of fraction bound for complexes in panel A. Mean and SEM values from triplicate EMSAs are plotted. (**C**) Superposition of crystal structures of ternary complexes of Pum and Nos with *hb* NRE2 RNA (yellow protein with orange RNA) and *CycB* NRE RNA (light blue protein and RNA). Nos is represented as a blue ribbon diagram. Zn atoms are shown as grey spheres. (**D**) Interaction of Pum with nucleotides 4–8 of *hb* NRE2 RNA ternary complex. (**E**) Interaction of Pum with nucleotides 4–7 of *CycB* NRE RNA within the Nos-Pum-*CycB* NRE RNA ternary complex.

The following figure supplements are available for figure 6:

**Figure supplement 1.** Nos promotes ternary complex formation with Pum and the *hb* NRE1 RNA.

**Figure supplement 2.** Nos promotes ternary complex formation with Pum and the *bcd* NRE RNA.

**Figure supplement 3.** Nos promotes ternary complex formation with Pum and the *hb* NRE2 +3G RNA.

**Figure supplement 4.** Nos and Pum do not bind the mutant *hb* NRE2 RNA.

**Figure supplement 5.** *hb* and *CycB* RNAs form different conformations in complex with Pum and Nos.

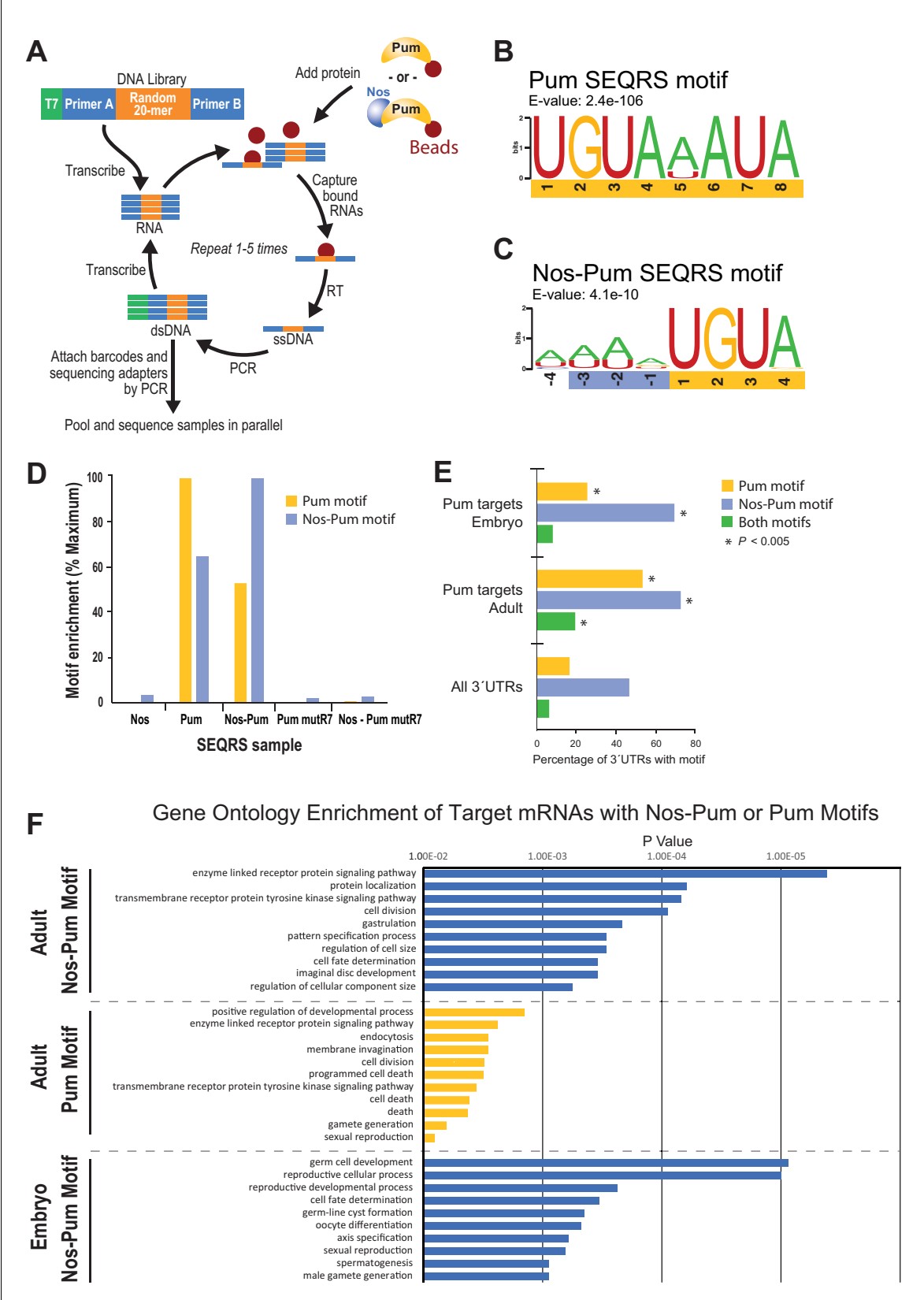

**Figure 7.** SEQRS analysis of Nos and Pum reveals specificity of RNA-binding activities. (A) Diagram of the SEQRS procedure. (B) Motif from SEQRS analysis of Pum. (C) Motif from SEQRS analysis of Nos-Pum complex. (D) The Nos-Pum and Pum motifs are preferentially enriched by their

*Figure 7 continued on next page*

*Figure 7 continued*

corresponding samples relative to three negative control conditions. These controls are Nos alone, an RNA-binding defective Pum mutR7, or Nos combined with Pum mutR7. Sequences are reported in *Figure 7—source data 2*. (E) Enrichment of Pum and Nos-Pum binding sites in 3′UTRs bound by Pum *in vivo* (*Gerber et al., 2006*), relative to 3′UTRs of all annotated *Drosophila* mRNAs. P values were determined using a chi-squared test. Test values are provided in *Figure 7—source data 2*. (F) Gene ontology enrichment of target mRNAs from *Drosophila* adults or embryos with Nos-Pum or Pum motifs. The ten most significantly enriched terms are shown for each category of target mRNA, ranked according to P-values. Complete tables of the gene ontology enrichment analysis is provide in *Figure 7—source data 1*.

The following source data and figure supplements are available for figure 7:

**Source data 1.** Related to *Figure 7F*.

**Source data 2.** Related to *Figure 7A–E*.

**Figure supplement 1.** Comparison of the reproducibility of two replicates of Pum SEQRS.

**Figure supplement 2.** Comparison of sequences selected in SEQRS for Pum and the Nos-Pum complex.

**Figure supplement 3.** Analysis of upstream nucleotides enriched in SEQRS by Nos in the ternary complex relative to Pum alone.

**Figure supplement 4.** Venn diagrams reveal differences in the extent of motif overlap in Pum bound transcripts in embryo (above) versus adult (below) (*Gerber et al., 2006*).

*Figure 7—figure supplement 4*). More Pum target mRNAs bear the Nos-Pum motif than the Pum motif, indicating that Nos expands the range of mRNAs regulated by Pum. Notably, few mRNAs have consensus PREs with upstream NBS motifs (*i.e.* both Nos-Pum and Pum motifs), such as *hb* mRNA. Most mRNAs with Pum motifs lack the upstream NBS, suggesting that they may be targeted by Pum alone or with other partners. Finally, we performed gene ontology analyses of target mRNAs bearing Pum or Nos-Pum motifs and observed that significantly enriched terms match the known functions of Nos and Pum, including body pattern formation and germline development, and also suggest new collaborative functions including regulation of cell division, receptor protein signaling, and cell fate determination (*Figure 7F*).

## Nanos expands the Pumilio target mRNA repertoire in cells

To evaluate the ability of Nos to enhance repression of mRNA targets with different NREs, we measured repression of reporter mRNAs containing the NREs for which we had examined ternary complex formation (*Figure 8A*). Because *hb* mRNA responds to the Nos concentration gradient in the *Drosophila* embryo, we varied the amounts of transfected Nos expression vector to produce different levels of Nos protein (*Figure 8B*). We found that Nos-enhanced repression of the 1x *hb* NRE2 reporter was dose dependent, increasing from 12% with 1 ng of transfected Nos expression vector to 68% with 200 ng (*Figure 8C*). Nos also elicited dose-dependent repression of reporters bearing the 1x *hb* NRE1 or *bcd* NRE, which were not stably bound by Pum alone, but supported ternary complex formation. In contrast, the Nos gradient did not enhance repression of reporters bearing the 1x *CycB* NRE or 1x *hb* NRE2 +3G mutant, relative to the background repression of a control reporter lacking a PRE (MCS) or the 1x *hb* NRE2 ACA mutant that abrogated Pum binding and ternary complex formation. Background repression of the *hb* NRE2 ACA reporter originating from binding to degenerate motifs in the RnLuc transcript may have limited the sensitivity of the 1x NRE reporters.

To improve the signal-to-noise ratio, we increased the number of NREs in each reporter to three adjacent sites, which dramatically increased cellular repression with lower amounts of Nos expression: 10 ng of transfected Nos expression vector resulted in 79% repression for the 3x *hb* NRE2 reporter (*Figure 8D*) versus 45% repression for the 1x *hb* NRE2 reporter (*Figure 8C*). Background repression of the 3x *hb* NRE2 ACA mutant remained low. Nos-enhanced repression of the 3x *CycB* NRE and mutant *hb* NRE2 +3G reporters was dose-dependent and significantly above background (*Figure 8D*). Thus, Nos enhances cellular repression of mRNAs with NREs that diverge from the PRE

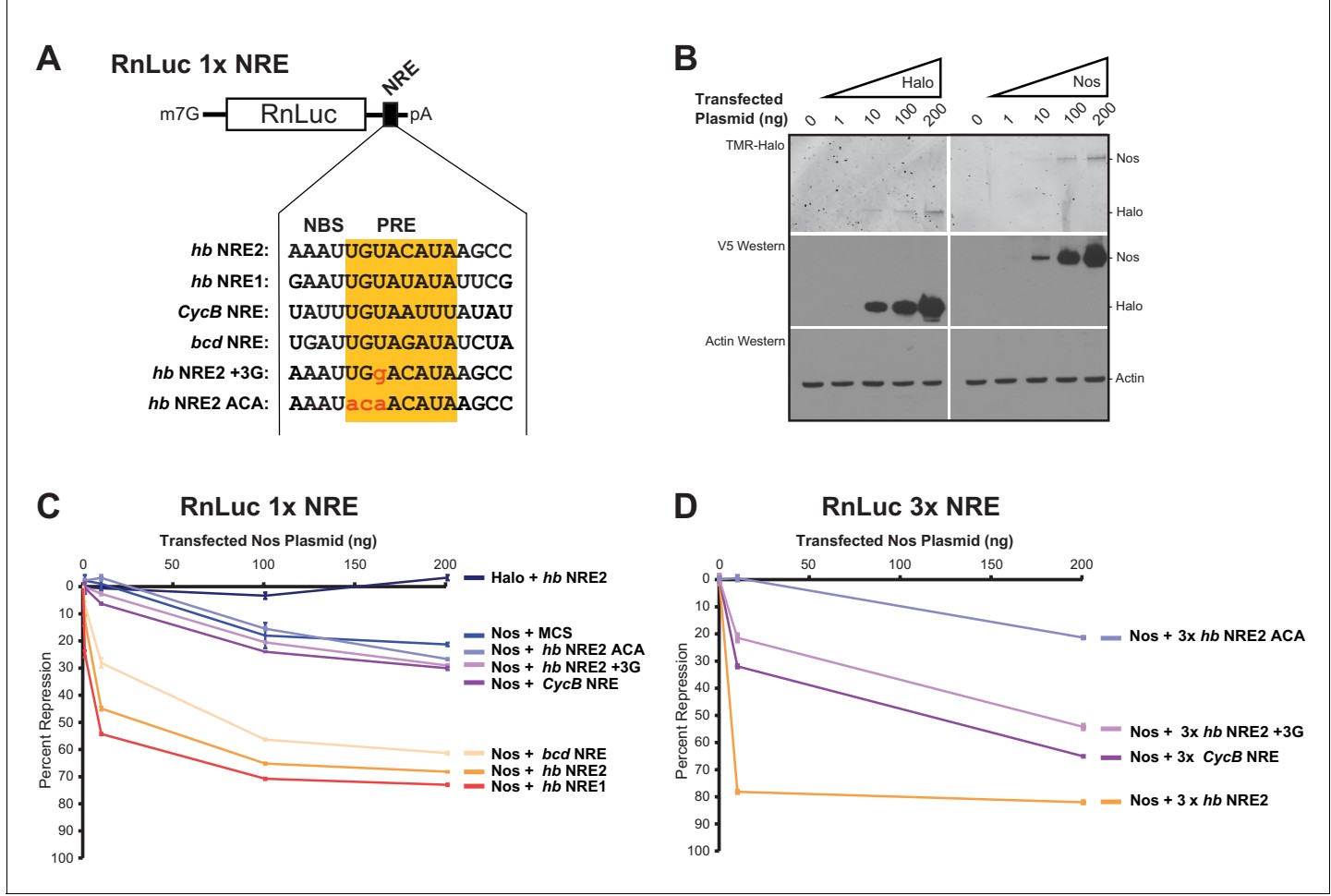

**Figure 8.** Nanos expands the Pumilio mRNA target repertoire in cells. (**A**) Diagram of the RnLuc 1x NRE reporters with minimal 3′UTRs containing WT or mutant (+3G and ACA) *hb* NRE2 sequences, the *hb* NRE1 sequence, or NRE sequences from the *Cyclin B (CycB)* and *bicoid (bcd)* mRNAs. The PRE sequence is yellow, whereas mutated positions are in lowercase red letters. (**B**) Dose-dependent expression of Halo-tag (Halo) and Halo-Nos in cells from transfected Nos plasmid was detected by fluorescent labeling with TMR (top panel) and western blotting with anti-V5 antibody (middle panel). As a loading control, actin was detected by western blotting (bottom panel). (**C**) Percent repression values are graphed for the variant RnLuc 1x NRE reporters with increasing amounts of transfected Nos or negative control Halo. (**D**) Percent repression values are graphed for the indicated RnLuc 3x NRE reporters with increasing amounts of transfected Nos expression plasmid. For Panels **C** and **D**, mean and SEM values from quadruplicate samples are shown. Statistical analysis of the data is reported in *Figure 8—source data 1*.

The following source data is available for figure 8:

**Source data 1.** Values and statistical analysis of luciferase reporter assays.

consensus and do not form stable complexes with Pum alone. Moreover, multiple weak NREs, such as 3x *CycB* NRE or 3x *hb* NRE2 +3G, confer substantial repression that approaches the level of a single strong NRE (e.g. 1x *hb* NRE2). As a consequence, Nos appears to expand the Pum target mRNA repertoire beyond those with perfect 5′-UGUAHAUA core PRE consensus sequences by stabilizing Nos-Pum-RNA complexes.

## Discussion

Hundreds of proteins that bind mRNAs in cells have now been identified, and large-scale efforts to find their specificities and mRNA targets are underway (*Baltz et al., 2012*; *Castello et al., 2012*; *Gerstberger et al., 2014*; *Sundararaman et al., 2016*). To understand their regulatory functions,

the emerging challenge is to classify these regulatory RBPs by their protein folds, specificities, *in vivo* target RNAs, and expression patterns. Furthermore, as the regulatory targets of individual RBPs are established, it will be crucial to examine overlapping networks to identify cooperative regulation by multiple RBPs. However, our current knowledge of how multiple RBPs collaboratively regulate an mRNA is limited. In this context, the mechanisms of combinatorial regulation by Nos and Pum revealed in this study, which visually capture decades of genetic and biochemical data, provide a foundation to derive general principles for how RBPs co-regulate target mRNAs.

Nos uses its three functional regions: N, Z, and C (*Figure 1*) to elicit repression of target mRNAs, and each region illustrates principles of combinatorial control. The Nos tandem ZFs form a module that adds protein-RNA contacts that strengthen RNA binding, using both side chain and main chain atoms to interact with the NBS. Nanos orthologs are found throughout Bilateria, and the tandem CCHC domains define the ZF-Nanos superfamily (PF05741) based on unique sequence, spacing and length compared to other ZFs. Interestingly, these ZFs were reported to be unique to Nos orthologs, and no structural homology to other RNA-binding ZFs was detected using a DALI structural search (*Hashimoto et al., 2010*; *Holm and Rosenström, 2010*). However, we manually compared the Nos ZFs with CCHC Zn knuckles (ZKs) from HIV nucleocapsid protein (HIVnc) (*De Guzman et al., 1998*) and found that Nos ZF2 is strikingly similar to the HIVnc ZKs (*Figure 9*).

Protein interactions between Nos and Pum further promote activity of the repression complex and, as revealed by the crystal structures, are mediated by the C-terminal region of Nos and repeats R7 and R8 of Pum. Incorporation of Nos into the Pum-RNA complex induced conformational changes in Pum that added additional RNA contacts. Protein interaction modules may also strengthen regulation by enhancing recruitment of cofactors. The N-terminal region of Nos, not included in our crystal structures, increases repression by the Nos-Pum complex by recruiting the CCR4-NOT deadenylase complex (*Kadyrova et al., 2007*; *Raisch et al., 2016*), which strengthens the independent repression activity of Pum (*Weidmann et al., 2012*, *2014*).

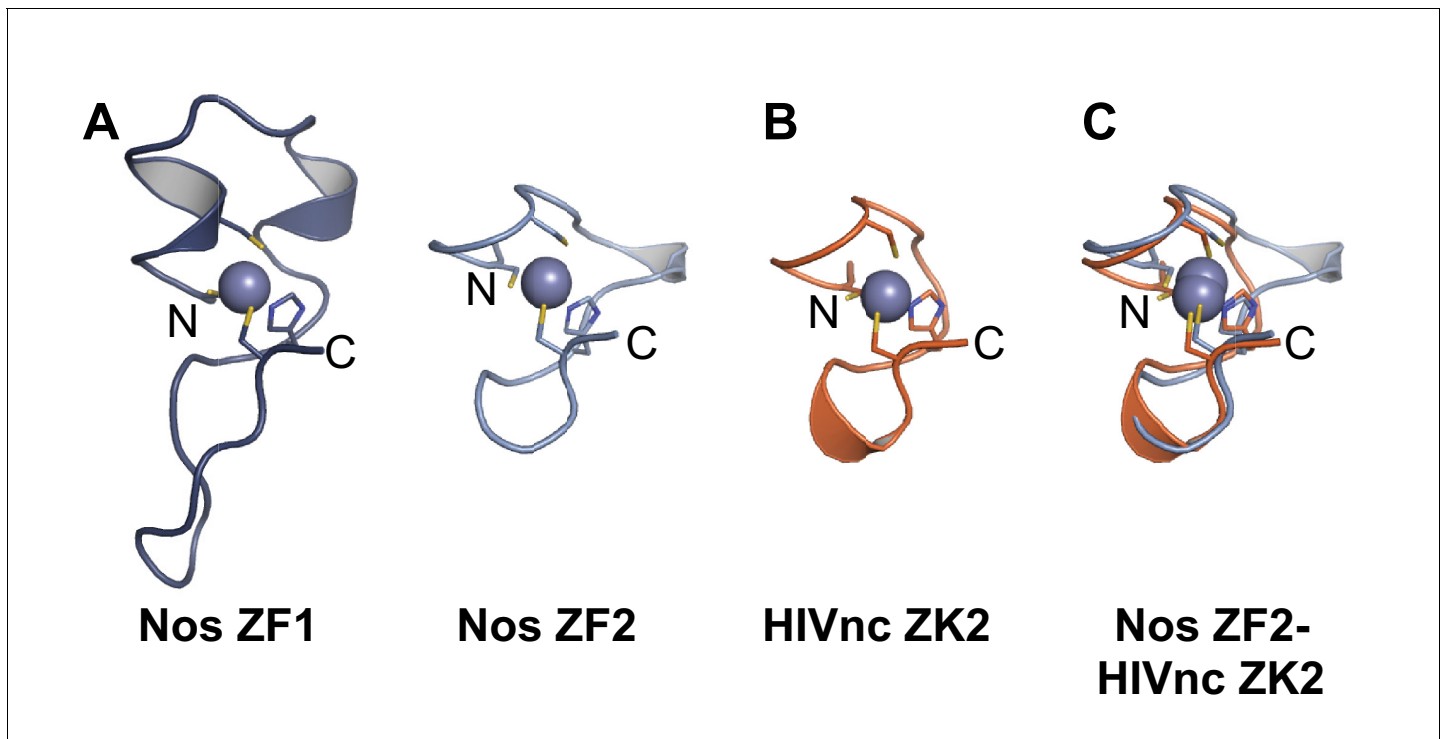

**Figure 9.** Nanos Zn finger 2 is structurally homologous to HIV nucleocapsid protein Zn knuckles. (**A**) Ribbon diagrams of Nos ZFs 1 and 2. The Nos ZFs follow the same structural topology, but ZF1 has longer loops than ZF2 that are N- and C-terminal to the Zn-coordinating histidine residue. (**B**) Ribbon diagram of HIV nucleocapsid (HIVnc) protein Zn knuckle (ZK) 2. The solution structures of HIV ZK1 and ZK2 are similar (rmsd 0.72 Å, 136 atoms). (**C**) Superposition of Nos ZF2 and HIVnc ZK2 (rmsd 1.2 Å, 88 atoms).

The functional importance of the interactions visualized by the Nos-Pum-RNA structures is corroborated by our findings and previous genetic and biochemical data. One of the first *nos* mutant alleles identified, *nos*[L7], causes a lethal loss of abdominal segments during embryogenesis and encodes a Nos protein with a deletion of residues 376–382 near the C terminus (*Arrizabalaga and Lehmann, 1999*; *Lehmann, 1988*). This deletion prevents formation of a Nos-Pum-RNA complex as assessed by yeast three-hybrid and in vitro pull-down assays (*Sonoda and Wharton, 1999*). Our crystal structures illustrate how contacts between the C-terminal region of Nos and Pum residues $F1367_{Pum}$ and $Q1337_{Pum}$ are lost with the *nos*[L7] deletion. A Pum mutation, F1367S, which blocks Pum and Nos association in a yeast 3-hybrid assay (*Edwards et al., 2001*) and impairs Nos enhancement of repression (*Weidmann and Goldstrohm, 2012*), also abrogated ternary complex formation in vitro (*Figure 4E*). Similarly, mutation of $Q1337_{Pum}$ prevented incorporation of Nos into the ternary complex (*Figure 4F*). In addition, $F1367_{Pum}$ is within the loop between Pum repeats R7 and R8 (*Figure 2—figure supplement 2*), explaining why insertion of four residues in the R7-R8 loop in the $Pum^{Mlu}$ mutant disrupts complex formation, *hb* repression and abdominal segmentation (*Sonoda and Wharton, 1999*; *Wharton et al., 1998*). These results show the crucial function of the interaction between the Nos C-terminal region and the convex surface presented by Pum repeats R7 and R8.

Based on our current understanding of the Pum-Nos interface and phylogenetic comparisons, the *Drosophila* Nos-Pum interaction is conserved among Dipteran orthologs, but it is not possible to predict conservation of the interaction in vertebrates. The Nos C-terminal region that contacts Pum is conserved within Diptera, but substantially diverges beyond that order (*Curtis et al., 1995*). Vertebrates have three Nos orthologs which share the tandem Nos ZFs but diverge substantially in their C termini. For example, the C termini of human Nos proteins are not homologous to each other or *Drosophila* Nos. The amino acid sequence of the 42-residue C-terminus of human NOS1 is enriched in prolines and is conserved throughout vertebrate NOS1 orthologs. The sequence of the 23-residue C-terminus of human NOS2 is enriched in arginines and is conserved among vertebrate NOS2 orthologs. Likewise, the sequence of the 63-residue C-terminus of human NOS3 is only shared by vertebrate NOS3 proteins. We also analyzed the conservation of the Nos-binding interface of Pum proteins and find that the contacts points observed in the Nos-Pum structure are conserved in Dipteran orthologs, but diverge throughout the tree of life. For example, residues equivalent to Pum Q1337 and F1367 change to proline and methionine, respectively, in vertebrate Pum homologs. In addition, vertebrate PUM proteins possess a three residue (GPH) insertion in the R7-R8 loop region that could augment protein contacts with vertebrate Nos proteins. Interestingly, deletion of these residues allows human Pum1 to interact with *Drosophila* Nos (*Sonoda and Wharton, 1999*).

Given these differences in Nos and Pum proteins, several possible evolutionary scenarios are worth consideration. First, the direct interaction between *Drosophila* Nos and Pum might be unique to Dipterans; however, Nos and Pum homologs from *C. elegans, Xenopus* and humans have been reported to interact and a simple loop deletion allows human Pum1 to interact with *Drosophila* Nos (*Jaruzelska et al., 2003*; *Kraemer et al., 1999*; *Lolicato et al., 2008*; *Nakahata et al., 2001*; *Sonoda and Wharton 1999*). Instead, the Nos and Pum contacts likely coevolved as the number of Nos and Pum homologs increased, perhaps restricting interactions between particular Nos and Pum proteins. Nos and Pum homologs may also interact indirectly, mediated by a bridging partner(s), as was suggested for *C. elegans* (*Kraemer et al., 1999*) and mouse (*Suzuki et al., 2016*).

We found that Nos stabilizes the ternary complex and adds 5′ sequence specificity. Nos specifically recognizes the 5′ sequence only in the context of the ternary complex and also induces a localized conformational change of Pum that adds contacts with the phosphate backbone of nucleotides -1 to -4. Previous reports indicated that the Nos tandem ZFs in isolation exhibit non-specific RNA binding (*Curtis et al., 1997*; *Hashimoto et al., 2010*). With our structure-based definition of the NBS, we find that the mutations in the RNAs tested by Curtis *et al.* were outside the NBS and thus would not have detected sequence-specific differences in Nos binding. With our findings, we can now attribute the negative effect of NBS mutations on abdominal segmentation *in vivo* (*Wharton et al., 1998*) and ternary complex formation in yeast three-hybrid assays (*Sonoda and Wharton, 1999*) to the loss of RNA recognition by Nos. Intriguingly, these newly identified Nos and Pum contacts with the 5′ NBS relax the sequence recognition requirements at the 3′ end of the PRE (*Figure 7C*), thereby allowing Nos and Pum to regulate mRNA targets bearing imperfect PREs, including *CycB* and *bcd*, that are not bound stably by Pum alone. As a result, the cooperative activity

of a second RBP (Nos) adds upstream RNA sequence specificity, which alters mRNA target selection of the primary RBP (Pum) and diversifies the range of target sequences.

Cooperative RNA recognition by Nos and Pum is reminiscent of cooperative binding of *msl2* RNA by Sex lethal (Sxl) and Upstream of N-Ras (Unr) (*Hennig et al., 2014*), yet the mechanisms and effects of cooperative recognition by Nos and Pum display novel distinguishing characteristics. Similar to Nos and Pum, Sxl and Unr proteins interact with each other only in the presence of target RNA, cooperatively interacting with a regulatory element that contains binding sites for each protein. Sxl and Unr recognize the RNA nucleotides between their individual binding sites, forming a sandwich or a "triple zipper" arrangement that creates a longer recognition element than either protein alone. Nos and Pum recognize their respective NBS and PRE elements; however, Nos recognition of the 5′ NBS and clamp-like binding to Pum and RNA relaxes the sequence requirements of Pum interaction with the 3′ end of the PRE, as with *CycB* mRNA, which effectively shifts the recognition sequence, rather than extending it. As a consequence, Nos and Pum cooperatively bind to RNAs that neither protein stably binds on its own.

Nos appears to dramatically alter the repertoire of transcripts bound by Pum. Our analysis of Pum target mRNAs using the SEQRS-derived Nos-Pum motif indicates that the majority of Pum target mRNAs contain the Nos-Pum motif but lack the full canonical PRE Pum recognition sequence. This suggests that, in addition to strengthening Pum repression activity, Nos can alter the identity of mRNAs regulated by Pum. Moreover, enrichment of specific gene ontology terms in the mRNAs bearing the Nos-Pum motif suggest that Nos can alter the biological functions controlled by Pum (*Figure 7*). These analyses allude to the utility of the combinatorial specificity *in vivo* and explain broadened regulation by Nos and Pum in *Drosophila* embryos and adults.

In addition to altering sequence specificity, our studies highlight other advantages of combinatorial RNA target regulation by Nos and Pum in vivo. First, repression is responsive to the Nos protein concentration. In the *Drosophila* embryo, the Nos protein gradient is highest at the posterior end where *hb* expression must be repressed for abdomen formation (*Barker et al., 1992*). We showed that higher Nos levels induced greater repression of reporters bearing *hb* NREs, mirroring the effect in the embryonic posterior. Interestingly, the expression of Nos is dynamic over the course of development and likely modulates Pum activity temporally as well as spatially (*Wang and Lehmann, 1991*). Second, repression is affected by the sequence of the NREs; reporters bearing imperfect PREs were also repressed, but required higher Nos protein levels for equivalent effect. Third, the number of Nos-Pum binding sites in the mRNA modulates regulatory activity. For example, *hb* mRNA bears two NREs, each with consensus NBS and PRE sites. We found that each site was highly responsive to Nos-enhanced, Pum-mediated repression, and multiple sites conferred greater responsiveness. Repression of the *CycB* PRE reporter required higher concentrations of transfected Nos, mirroring the requirement for concentrated Nos protein in pole cells where *CycB* translation is repressed (*Asaoka-Taguchi et al., 1999*; *Kadyrova et al., 2007*). Inclusion of three *CycB* PREs supported repression at lower Nos concentrations, consistent with previous studies showing how the number and quality of PREs in the *hb* 3′UTR, and the amount of Nos protein expressed, confer the precise level of regulation upon the *hb* transcript in embryonic development (*Wharton and Struhl, 1991*). Indeed, although the *CycB* mRNA 3′ UTR contains no perfect PRE, it does contain 7 sequence elements with an NBS upstream of a partial PRE with the 5′ UGU sequence. Together, these results define multiple parameters that contribute to biologically relevant levels of regulatory activity.

Other RBPs collaborate with Pum to regulate mRNAs. For example, the Brain Tumor (Brat) protein contributes to repression of *hb* expression in the embryo by recognizing an element located upstream of the NBS and PRE in each NRE (i.e. the so-called Box A motif) (*Laver et al., 2015*; *Loedige et al., 2014*; *Sonoda and Wharton, 2001*). Moreover, additional partner proteins can modulate RNA binding of PUF proteins (*Campbell et al., 2012a*, *2012b*; *Menichelli et al., 2013*) and the Nos-Pum-RNA ternary complex suggests common mechanisms of co-regulation. For instance, RNA-binding affinity and specificity of the *C. elegans* PUF protein, FBF-2, is modulated by interactions with CPB-1 (Cytoplasmic Polyadenylation element Binding protein 1) (*Menichelli et al., 2013*). Intriguingly, CPB-1 and other partners bind to a region of FBF-2 that corresponds to the R7-R8 loop of Pum that binds Nos (*Campbell et al., 2012b*; *Menichelli et al., 2013*; *Wu et al., 2013*). Thus, the Nos-Pum-RNA complex exemplifies a protein-protein interaction hotspot (*Campbell et al., 2012b*). Our results indicate that formation of PUF protein-partner complexes can alter the PUF protein

conformation to create additional protein-RNA contacts that strengthen RNA binding while modulating target specificity and regulation.

In conclusion, the partnership of Nos and Pum illustrates the profound influence combinatorial control can have on gene expression, demonstrating how location-specific regulation is achieved through the action of RBPs with different spatial distributions, how the addition of a second RBP shifts the RNA recognition motif of the first RBP to modulate target selection, and how regulatory sensitivity can be adjusted by the number and quality of binding sites within a target mRNA and by RBP expression levels. This paradigm emphasizes the importance of understanding how the multitude of RNA-binding factors collaborate to control mRNA stability, translation, processing, and localization.

## Materials and methods

### Plasmids

Reporter plasmids pAc5.1 FFluc, pAc5.1 RnLuc *hb* 3′UTR were previously described (*Weidmann and Goldstrohm, 2012*, *Weidmann et al., 2014*), as was the control plasmid pIZ Halo-tag. The pIZ Halo-Nos expression vector (NZC) was created by inserting the *Drosophila* Nanos cDNA (NP_001262723.1) into the XbaI site of pIZ Halo-tag, which contains an N-terminal Halo-tag with a TEV protease cleavage site and a C-terminal V5 epitope. The Nos sequence was amplified from whole fly cDNA and corresponds to isoform Nos-PB, which lacks an alternatively spliced exon encoding a 19 amino acid sequence aa14-VGVANPPSLAQSGKIFQLQ-32 present in the N-terminus of Nos-PA (NP_476658.1). For consistency with the originally identified domain boundaries and mutants, the reported amino acid positions correspond to Nos-PA (e.g. C319Y and C354Y correspond to C300Y and C335Y of Nos-BP). Using the Nos plasmid as a template, the C319Y, C354Y, I376A, M378A, and I382A mutations were generated via QuikChange site-directed mutagenesis (QC-SDM, Agilent). The following mutations and truncations were created using inverse PCR from the Nos plasmid template: F321A, N325A, Y352A, T366A, K368Q, Y369A, NZ (aa1-373), ZC (aa289-401), N (aa1-294), Z (aa289-373), C (aa374-401), Δ376–382 (aa1-375 + 383–401), Δ383–393 (aa1-382 + 394–401), and Δ394–401 (aa1-393).

Bacterial expression and purification of recombinant Pum or Nos for EMSAs was achieved using pFN18K (Promega) with an N-terminal Halo-tag and a TEV cleavage site. The Pum RNA-binding domain sequence, encoding aa1091-1426 of NP_001262403.1, including an N-terminal triple FLAG tag was inserted into pFN18K to create pFN18K Pum plasmid. QC-SDM was used to generate the F1367S Pum mutant and the RNA-binding defective mutR7 (wherein RNA recognition amino acids are mutated: S1342A N1343A E1346A) plasmids. Inverse PCR was used to create Pum Q1337A from the wild type template. The same strategy was applied to generate pFN18K NosZC and the C319Y and C354Y mutant vectors with appended C-terminal V5 epitopes. For crystallographic studies, *Drosophila* Pum RNA-binding domain (amino acids 1091–1426) and Nanos ZF domain (amino acids 289–401) were subcloned into the pSMT3 vector with an N-terminal His$_6$-SUMO tag (kindly provided by Christopher Lima, Memorial Sloan Kettering Cancer Center, New York).

Reporters used for Nos enhancement of Pum repression in *Figure 8* were made in the pAc5.4 vector, wherein a cryptic cleavage/polyadenylation element intrinsic to pAc5.1 vector was removed and a degenerate PRE motif in the RnLuc ORF was inactivated by introducing synonymous codons. Complementary DNA oligos (IDT) bearing wild type and mutant NREs, listed below, were inserted into XhoI and Not1 restriction sites within the 3′UTR. The 3x *hb* NRE2, 3x *CycB* NRE, 3x *hb* NRE2 +3G, and 3x *hb* NRE2 ACA reporters were generated in an identical fashion using oligos with three repeated NRE elements.

### Northern analysis

RNA was purified from 2 million D.mel-2 cells transfected with the plasmids indicated in *Figure 1—figure supplement 1*. Cells were harvested at 1000 × g for 3 min, washed twice in PBS, and RNA was purified from cell pellets using TRIzol reagent (Life Technologies). Total RNA preparations were then analyzed by Northern blotting as previously described (*Blewett and Goldstrohm, 2012*). RNA was separated in a denaturing 0.85% agarose gel containing 1x MOPS and formaldehyde. RNA was transferred by blotting to an Immobilon NY+ membrane (Millipore). Membranes were then

UV-crosslinked and probed for the RNAs indicated in the figure. For RnLuc reporter, a [32]P body-labeled, antisense RNA probe was created by in vitro transcription. The following primers were used to amplify templates for creation of RnLuc RNA probes. The T7 promoter sequence is underlined and gene specific regions are bolded.

RnLuc 3′ forward primer: 5′-**GGGCGAGGTTAGACGGCCTACCCT**

RnLuc 3′ reverse primer: 5′-GGATCCTAATACGACTCACTATAGG**GCGGCCAGCGGCCTTGG**

The 7SL RNA was detected on northern blots using a [32]P 5′ end-labeled DNA oligo with the following sequence.

7SL Probe: 5′-CACCCCTGGCCCGGTTCATCCCTCCTTAGCCAACCTGAATGCCACGG.

The radioactive blots were exposed to a storage phosphor screen. The signal on the screen was captured with a Typhoon Trio imager (GE Healthcare) and subsequently quantified using Image-Quant TL Software (GE Healthcare). Each RnLuc signal was normalized to the 7SL signal. Statistical analysis of Northern blot data from three replicate cell cultures is reported in *Figure 1—figure supplement 1—source data 1*.

## Oligonucleotides

The oligos used to create RnLuc 1xNRE reporter plasmids used for Nos enhancement of Pum repression are as follows (Restriction site overhangs are indicated in bold, PRE sequences underlined, mutations lowercase):

*hb* NRE2 Forward: 5′-**TCGA**CGAAAATTGTACATAAGCC

*hb* NRE2 Reverse: 5′-**GGCC**GGCTTATGTACAATTTTCG

*hb* NRE2 +3G Forward: 5′-**TCGA**CGAAAATTGgACATAAGCC

*hb* NRE2 +3G Reverse: 5′-**GGCC**GGCTTATGTcCAATTTTCG

*hb* NRE2 ACA Forward: 5′-**TCGA**CGAAAAT**aca**ACATAAGCC

*hb* NRE2 ACA Reverse: 5′-**GGCC**GGCTTATGT**tgt**ATTTTCG

*hb* NRE1 Forward: 5′-**TCGA**CCAGAATTGTATATATTCG

*hb* NRE1 Reverse: 5′-**GGCC**CGAATATATACAATTCTG

*bcd* NRE Forward: 5′-**TCGA**AAGTGATTGTAGATATCTA

*bcd* NRE Reverse: 5′-**GGCC**TAGATATCTACAATCACTT

*CycB* NRE Forward: 5′-**TCGA**GACTATTTGTAATTTATATC

*CycB* NRE Reverse: 5′-**GGCC**GATATAAATTACAAATAGTC

*hb* NRE2 -1C Forward: 5′-**TCGA**CGAAAA**c**TGTACATAAGCC

*hb* NRE2 -1C Reverse: 5′-**GGCC**GGCTTATGTACA**g**TTTTCG

*hb* NRE2 -2C Forward: 5′-**TCGA**CGAAA**c**TTGTACATAAGCC

*hb* NRE2 -2C Reverse: 5′-**GGCC**GGCTTATGTACAA**g**TTTCG

Synthetic RNAs (IDT) used in EMSA experiments include the following (with PRE elements underlined and mutations in lowercase bold):

Cy5- *hb* NRE2 RNA: 5′-Cy5-rUUGUUGUCGAAAAUUGUACAUAAGCC.

*hb* NRE2 RNA: 5′-rAAAUUGUACAUAAGCC

*hb* NRE2 +3G RNA: 5′-rAAAUUGgACAUAAGCC

*hb* NRE2 ACA RNA: 5′-rAAAU**aca**ACAUAAGCC

*hb* NRE1 RNA: 5′-rGAAUUGUAUAUAUUCG

*bcd* NRE RNA: 5′-rUGAUUGUAGAUAUCUA

*CycB* NRE RNA: 5′-rUAUUUGUAAUUUAUAUC

*hb* NRE2 -1C RNA: 5′-rAAA**c**UGUACAUAAGCC

*hb* NRE2 -2C RNA: 5′-rAA**c**UUGUACAUAAGCC

## Cell culture and transfections

D.mel-2 cells were cultured at 28°C in Sf-900 III SFM (Life Technologies) with 50 Units/mL penicillin and 50 μg/mL streptomycin. Transfections were performed as described (*Weidmann and Gold-strohm, 2012*; *Weidmann et al., 2014*) using Effectene (Qiagen). Each transfected well of a 6-well plate contained 5 ng Firefly Luciferase internal control plasmid (pAc5.1 FFLuc), 10 ng of the indicated RnLuc reporter plasmid, and 200 ng total of protein expression vector, 43–44 μl of EC buffer, 1.6 μl of enhancer, 2 μl of Effectene, 300 μl of new Sf-900 III SFM, and 1.6 mL of D.mel-2 cells (1 × 10^6 cells/mL). For Nos experiments, 10 ng of pIZ Nos expression vector (unless otherwise noted)

was balanced with empty pIZ vector for a total mass of 200 ng. For each Nos transfection gradient, pIZ was also used to balance the total mass of transfected expression vector to 200 ng. Transfection conditions for the experiment in *Figure 4D* differed in the following manner: each well of a 6-well plate contained 5 ng Firefly Luciferase internal control plasmid (pAc5.4 FFLuc), 10 ng of the reporter gene pAc5.4 RnLuc *hb* 3′UTR, and 100 ng of the indicated protein expression plasmid. Total transfected DNA was set at 3 µg per well, balanced by pIZ plasmid. Fugene HD (Promega) transfection reagent was used at a 4:1 ratio (µl Fugene HD: µg of DNA), prepared in 150 µL Sf-900 III SFM media and incubated for 15 min at room temperature prior to application to $2x10^6$ cells in 2 ml of media.

## Luciferase assays

Luciferase reporter assays were performed using the Dual-Glo assay (Promega) and a GloMax Discover luminometer. A relative response ratio (RRR) was calculated from RnLuc/FFLuc signals for each sample. Percent repression values were calculated as previously described (*Van Etten et al., 2013*; *Weidmann and Goldstrohm, 2012*). The pIZ-Halo-tag vector served as the negative control for Halo-NZC Nos constructs. For the Nanos gradient experiments in *Figure 8*, the 0 ng condition, corresponding to 200 ng of transfected pIZ plasmid, served as the negative control. Data and statistical analyses of all reporter assays are reported in *Figure 1—source data 1*, *Figure 4—source data 1*, *Figure 5—source data 1*, and *Figure 8—source data 1*. Four replicate cell cultures were analyzed in each experiment as indicated in the figure legends. Results were validated in multiple independent experiments performed on different days.

## Protein expression and purification for RNA-binding assays

Recombinant Pum and Nos for EMSAs were expressed in KRX *E. coli* cells (Promega) in 2xYT media with 25 µg/mL Kanamycin and 2 mM $MgSO_4$ at 37°C to $OD_{600}$ of 0.7–0.9, at which point protein expression was induced with 0.1% (w/v) rhamnose for 3 hr. Cell pellets were washed with 50 mM Tris-HCl, pH 8.0, 10% [w/v] sucrose and pelleted again. Pellets were suspended in 25 mL of 50 mM Tris-HCl pH 8.0, 0.5 mM EDTA, 2 mM $MgCl_2$, 150 mM NaCl, 1 mM DTT, 0.05% (v/v) Igepal CA-630, 1 mM PMSF, 10 µg/ml aprotinin, 10 µg/ml pepstatin, and 10 µg/ml leupeptin. To lyse cells, lysozyme was added to a final concentration of 0.5 mg/mL and cells were incubated at 4°C for 30 min with gentle rocking. $MgCl_2$ was increased to 7 mM and DNase I (Roche) was added to 10 µg/mL followed by incubation for 20 min. Lysates were cleared at 50,000xg for 30 min at 4°C. Halo-tag containing proteins were purified using HaloLink Resin (Promega) at 4°C. Beads were washed 3 times with 50 mM Tris-HCl pH 8.0, 0.5 mM EDTA, 2 mM $MgCl_2$, 1 M NaCl, 1 mM DTT, 0.5% [v/v] Igepal CA-630) and 3 times with Elution Buffer (50 mM Tris-HCl, pH 7.6, 150 mM NaCl, 1 mM DTT, 20% [v/v] glycerol). After washing, beads were resuspended in Elution Buffer with 30 U of AcTEV protease (Invitrogen), cleavage proceeded for 24 hr at 4°C, and beads were removed by centrifugation through a micro-spin column (Bio-Rad).

## Electrophoretic mobility shift assays

All RNA-binding reactions were performed in RNA-binding Buffer (50 mM Tris-HCl pH 7.6, 150 mM NaCl, 2 mM DTT, 2 µg/mL BSA, 0.01% [v/v] Igepal CA-630, 0.02% bromophenol blue, 20% [v/v] glycerol). Reactions were equilibrated for 3 hr at 4°C. 5% native polyacrylamide TBE mini-PROTEAN gels (Bio-Rad) were pre-run for 3 hr at 50V before loading 5 µl of each sample and then run at 50V for 2–2.25 hr at 4°C.

For EMSAs with fluorescently-labeled *hb* NRE2 RNA, reactions contained 1 nM target RNA and the concentrations of PUM-HD and NosZC are as noted in *Figure 3B*. For $K_d$ measurements shown in *Figure 3C and D*, Pum was held constant at 100 nM, and NosZC concentrations included 0, 0.5, 1, 2, 5, 10, 20, 50, 100, 200, 500, and 1000 nM. In the RNA + NosZC lane, RNA was at 1 nM and NosZC was at 1000 nM.

For $K_d$ measurements with radioactive RNA oligos, reactions contained 0.3 nM RNA that were labeled at their 5′ ends using T4 Kinase (NEB) with ATP [γ-$^{32}$P] (Perkin-Elmer). Where indicated for *hb* NRE2 RNA, the concentration of NosZC was held constant at 400 nM while Pum was titration included concentrations of 0, 0.2, 0.5, 1, 2, 5, 10, 20, 50, 75, 100, 150, and 200 nM. For all other RNAs tested in the presence of Nos, the reactions contained 1 µM NosZC while Pum concentration was titrated from 0, 2, 5, 10, 20, 50, 100, 200, 300, 400, 500, 750, and 1000 nM. Gels containing

radioactive RNAs were dried onto Whatmann filter paper. The radioactive gels were then exposed to a storage phosphor screen (GE) for 16 hr.

EMSAs were imaged with a Typhoon Trio imager (GE Healthcare) and subsequently quantified using ImageQuant TL software. Fraction bound values from three replicate EMSAs were plotted against titrated protein concentration, and $K_d$ was calculated via nonlinear regression analysis for one site interaction with GraphPad Prism software (GraphPad Software, Inc.).

## Protein expression and purification for crystallization

Pum and Nos proteins (pSMT3 vectors) were overexpressed in *E. coli* strain BL21-CodonPlus (DE3)-RIL (Agilent) at 16°C for 20 hr after induction with 0.2 mM IPTG. The cell pellet was resuspended in lysis buffer containing 20 mM Tris (pH 8.0), 0.5 M NaCl, 20 mM imidazole, 5% (v/v) glycerol, and 0.1% (v/v) β-mercaptoethanol. Both proteins were purified by a similar procedure involving four sequential steps: a Ni-NTA chelating column, Ulp1 overnight incubation to remove the His$_6$-SUMO tag, a heparin column and lastly a Superdex 75 column (GE Healthcare). The Ni elution buffer contained 20 mM Tris (pH 8.0), 50 mM NaCl, 0.2 M imidazole, and 1 mM DTT. A gradient elution was run on the heparin column with buffer A containing 20 mM Tris (pH 8.0) and 1 mM DTT and buffer B containing additional 1 M NaCl. The Superdex 75 column buffer contains 20 mM Hepes (pH 7.4), 0.15 M NaCl, and 2 mM DTT. For the ternary complex formation, a 16-nt *hb* NRE2 RNA (5′-AAAU<u>U-GUACAUA</u>AGCC) and a 14-nt *CycB* NRE RNA (5′-UAUU<u>UGUAAUUUA</u>U) were used, separately. Purified Pum and Nos were concentrated and mixed together with RNA in a molar ratio of 1:1:1.1. After overnight incubation at 4°C, the mixture was loaded onto a Superdex 200 10/300 GL column. Peak fractions containing ternary complexes were concentrated to OD$_{280}$ ~ 7 in a buffer of 20 mM HEPES (pH 7.4), 0.15 M NaCl, and 2 mM DTT for crystallization. For the Pum-RNA binary complex, concentrated Pum (OD$_{280}$~4.0) was incubated with 8-nt *hb* PRE2 RNA (5′-<u>UGUACAUA</u>) in a molar ratio of 1:1.2 on ice for 2 hr. The mixture was directly put into crystallization trays. Binary complexes were also formed with 12-nt *hb* NRE2 RNA (5′-AAAU<u>UGUACAUA</u>), but no crystals were obtained.

## Crystallization and data collection

Crystals of Nos-Pum-*hb* NRE2 RNA complex were obtained by hanging drop vapor diffusion, mixing 2 μL of sample and 2 μL of reservoir solution [1.1 M (NH$_4$)$_2$SO$_4$, 0.1 M MES, pH 5.6] at 20°C. Crystals of Pum-Nos-*CycB* NRE RNA complex were obtained in the condition of 0.9 M (NH$_4$)$_2$SO$_4$, 0.1 M MES, pH 5.6 by microseeding using the *hb* complex crystals. Crystals of Pum-*hb* PRE2 RNA binary complex were obtained in the condition of 15% (w/v) PEG 3350, 0.1 M bis-tris, pH 6.5 and 0.2 M NH$_4$OAc. All crystals were transferred to a cryo-solution containing the reservoir solution with additional 15–20% (v/v) glycerol, and flash frozen in liquid nitrogen. X-ray diffraction data were collected at the SER-CAT Beamline 22-ID at the Advanced Photon Source, Argonne National Laboratories.

## Structure determination

Crystals of ternary complexes belong to P6$_5$22 space group with one complex in an asymmetric unit. The structure of Pum-Nos-*hb* NRE2 RNA complex was determined by molecular replacement using the structures of *Drosophila* Pum (PDB: 3H3D) and zebrafish Nanos ZF domain (PDB: 3ALR) as the search model with Phaser. Iterative model building was done with COOT and Phenix (*Adams et al., 2002*; *Emsley and Cowtan, 2004*). The structure of Pum-Nos-*CycB* NRE RNA complex was determined by molecular replacement using the *hb* complex as the search model. Residues 1092–1419 in Pum, residues 316–385 in Nos and RNA bases from −4 to +8 are modeled in the *hb* NRE2 structure. Residues 1092–1102 and 1121–1418 in Pum, residues 316–384 in Nos, and RNA bases from −4 to +7 are modeled in the *CycB* NRE structure.

Crystals of Pum-*hb* PRE2 RNA binary complex belong to C2 space group with one complex in an asymmetric unit. The structure was determined by molecular replacement using the structure of *Drosophila* Pum (PDB: 3H3D) as the search model. Residues 1090–1426 in Pum and RNA bases +1 to +8 of PRE2 are modeled in the binary structure. Data collection and refinement statistics are presented in *Table 1*.

## Western blotting

Western blotting from luciferase assay samples was performed as previously described (*Weidmann and Goldstrohm, 2012*, *Weidmann et al., 2014*). Where indicated, proteins were detected using a V5 monoclonal antibody (Invitrogen) and horseradish peroxidase conjugated goat anti-mouse IgG (Thermo Scientific). Signals were detected using either Pierce ECL Western blotting substrate (Thermo) or Immobilon western chemiluminescent substrate (Millipore) and autoradiography film.

## Fluorescent detection of Halo-tag fusion proteins

Protein extracts of D.mel-2 cells expressing Halo-tag fusions were prepared as previously described (*Weidmann and Goldstrohm, 2012*). Extracts were then incubated with 100 nM Halo-tag TMR Ligand (Promega) for 30 min on ice, protected from light. After labeling, extracts were separated via SDS polyacrylamide gel electrophoresis and labeled proteins were imaged with a Typhoon Trio imager (GE Healthcare).

## SEQRS

SEQRS was conducted as described with minor modifications (*Campbell et al., 2014*) on the following samples: 1) wild type Pum alone, 2) wild type Nos alone, 3) wild type Pum with Nos, 4) Pum mut R7 alone and 5) Pum mutR7 with Nos. The proteins were purified as described above for EMSA experiments except that Magnetic Halolink beads (Promega) were used and the Pum test proteins remained covalently bound via N-terminal Halotag to the beads. For Nos alone (sample 2), Nos remained linked to the magnetic beads. For the other samples that contained Nos (samples 3 and 5), Nos protein was cleaved from the beads using TEV protease and equimolar amount was added to the Pum-linked beads. The initial RNA library was transcribed from 1 μg of input dsDNA using the AmpliScribe T7-Flash Transcription Kit (Epicentre). 200 ng of DNase treated RNA library was added to 100 nM of Halo-tagged proteins immobilized onto magnetic resin (Promega). The volume of each binding reaction was 100 μl in SEQRS buffer containing 200 ng yeast tRNA competitor and 0.1 units of RNase inhibitor (Promega). The samples were incubated for 30 min at 22°C prior to magnetic capture of the protein-RNA complex. The binding reaction was aspirated and the beads were washed four times with 200 μl of ice cold SEQRS buffer. After the final wash step, resin was suspended in elution buffer (1 mM Tris pH 8.0) containing 10 pmol of the reverse transcription primer. Samples were heated to 65°C for 10 min and then cooled on ice. A 5 μl aliquot of the sample was added to a 10 μl ImProm-II reverse transcription reaction (Promega). The ssDNA product was used as a template for 25 cycles of PCR using a 50 μl GoTaq reaction (Promega). Sequencing data were processed as described (*Campbell et al., 2012a*). Sequence logos corresponding to consensus binding motifs were generated by MEME analysis of the 100 most-enriched sequences (reported in *Figure 7— source data 2*) (*Bailey et al., 2006*). Enrichment of Pum and Nos-Pum binding sites in 3′UTRs was analyzed for all mRNAs and for mRNAs bound by Pum *in vivo* using the dataset from *Gerber et al., 2006*. Pattern matching to the Pum and Nos-Pum motifs reported in *Figure 7* were preformed using the grep Perl function from command line. Significance values compared to all 3′UTR sequences was determined using chi-squares test using GraphPad Prism (reported in *Figure 7—source data 2*). Gene ontology enrichment analysis was performed using DAVID (*Huang et al., 2009*, *2008*) and Venn diagrams were generated using Venn Diagram Plotter (Pacific Northwest National Laboratory).

## Acknowledgements

We thank L Pedersen and the staff of the Southeast Regional Collaborative Access Team beamlines for assistance with X-ray data collection and Drs. P Blackshear, K McCann, R Trievel, and P Freddolino for comments on the manuscript. This research was supported in part by NIGMS R01GM105707 (ACG); NIH NRSA 5T32GM007544 (CW. and RA.); a Research Scholar Grant, RSG-13-080-01-RMC, from the American Cancer Society (ACG); a Graduate Research Fellowship, DGE 1256260, from NSF (RMA.) and the Intramural Research Program of the National Institutes of Health, National Institute of Environmental Health Sciences (TMTH). The Advanced Photon Source used for this study is supported by the US Department of Energy, Office of Science, Office of Basic Energy Sciences, under contract no. W-31-109-Eng-38.

## Additional information

### Funding

| Funder | Grant reference number | Author |
|---|---|---|
| National Institute of General Medical Sciences | R01GM105707 | Chase A Weidmann<br>René M Arvola<br>Jordan Killingsworth<br>Aaron C Goldstrohm |
| National Institutes of Health | NRSA 5T32GM007544 | Chase A Weidmann<br>René M Arvola |
| American Cancer Society | RSG-13-080-01-RMC | Chase A Weidmann<br>Aaron C Goldstrohm |
| National Institute of Environmental Health Sciences | Intramural Research Program | Chen Qiu<br>Traci M Tanaka Hall |
| U.S. Department of Energy | W-31-109-Eng-38 | Chen Qiu<br>Traci M Tanaka Hall |
| National Science Foundation | DGE 1256260 | René M Arvola |

The funders had no role in study design, data collection and interpretation, or the decision to submit the work for publication.

### Author contributions

CAW, CQ, RMA, ZTC, TMTH, Conception and design, Acquisition of data, Analysis and interpretation of data, Drafting or revising the article; T-FL, Acquisition of data, Analysis and interpretation of data; JK, Acquisition of data, Contributed unpublished essential data or reagents; ACG, Conception and design, Analysis and interpretation of data, Drafting or revising the article

### Author ORCIDs

Aaron C Goldstrohm, http://orcid.org/0000-0002-1867-8763

## Additional files

### Major datasets

The following datasets were generated:

| Author(s) | Year | Dataset title | Dataset URL | Database, license, and accessibility information |
|---|---|---|---|---|
| Qiu C, Hall TMT | 2016 | Crystal structure of the drosophila Pumilio RNA-binding domain in complex with hunchback RNA | http://www.rcsb.org/pdb/search/structid-Search.do?structureId=5KLA | Publicly available at the RCSB Protein Data Bank (accession no: 5KLA) |
| Qiu C, Hall TMT | 2016 | Crystal structure of the Pumilio-Nos-hunchback RNA complex | http://www.rcsb.org/pdb/search/structid-Search.do?structureId=5KL1 | Publicly available at the RCSB Protein Data Bank (accession no: 5KL1) |
| Qiu C, Hall TMT | 2016 | Crystal structure of the Pumilio-Nos-CyclinB RNA complex | http://www.rcsb.org/pdb/search/structid-Search.do?structureId=5KL8 | Publicly available at the RCSB Protein Data Bank (accession no: 5KL8) |

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
