## [Decision Letter]

Thank you for submitting your article "*Drosophila* Nanos acts as a molecular clamp that modulates the RNA- binding and repression activities of Pumilio" for consideration by *eLife*. Your article has been reviewed by three peer reviewers, one of whom is a member of our Board of Reviewing Editors, and the evaluation has been overseen by James Manley as the Senior Editor. The reviewers have opted to remain anonymous.

The reviewers have discussed the reviews with one another and the Reviewing Editor has drafted this decision to help you prepare a revised submission.

The article by Weidman and colleagues addresses a long-lasting question regarding how the RNA binding proteins Pumilio and Nanos regulate their targets. While the structures of Pumilio bound to RNA targets had been identified a while ago, it remained unclear how Nanos may regulate target specificity for a Pumilio-Nanos-NRE tertiary complex. In a beautiful set of experiments connecting in vitro binding and translational repression studies with structural analysis these questions are addressed in the present paper. The analysis is thorough and the data are well controlled and illustrated. The authors are able to identify a new sequence upstream of the Pumilio binding sites important for Nanos contact with RNA in the presence of Pumilio. It also represents one of the few examples of how two RBP cooperate to achieve sequence specificity and how one RBP (nanos) can modulate the sequence-specificity of the other one Pumilio. While this study does not include*in vivo* tests of the predicted interactions, several predictions have already been born out by previously published results. For example, *in vivo* studies had implicated sequences upstream of the canonical Pumilio binding sites in hb RNA regulation. Similarly, the new data illustrate the importance of the Nanos C-terminus in 'clamping' Pumilio to RNA, which was so far mostly implicated by genetics, for Nanos function in hb regulation and less so in germ line development.

Though the structures and binding studies nicely illustrate how cooperativity works in this context, it would have been really worthwhile for authors to use this term early on and would not wait till the middle of their discussion.

In addition, the paper would be improved if the interactions found

were discussed in terms of an evolutionary comparison. How conserved are these contacts? Can one use the structures to predict cooperativity of two RBPs in other organisms? This is not touched upon and would be very valuable.

Several issues should be clarified:

Much of the analysis is a comparison between the binary Pum-RNA complex and the ternary Nos-Pum-RNA complex. The structure of Pum changes in the region where Nos binds to it and to the RNA element. However, in the binary complex, the RNA doesn't really reach that area. It is shorter and contains only the PRE. Therefore, would Pum assume this altered conformation in the presence of a longer RNA without Nos? Or might it be closer to the conformation it assumes with Nos? This is an important point and bears on the conclusions of this paper.

It is very hard to see from the various figures, what is moving in Pum between the binary and ternary complexes. Perhaps a side by side of that area would be easier to follow.

The deletions that are discussed on Results section, subsection “Nanos increases the binding affinity of Pumilio for hunchback RNA”, that affect repression involve cutting out a few residues sometimes in the middle of a helix and it's not clear how well the structure would retain its integrity. The extent of repression might correlate with what percentage of folded protein they have in this region. Does this correlate with Pum-RNA binding?

The authors show a difference in the RNA structure between the hb and CycB complexes. However, the resolution is fairly low. How confident are they? What do the electron density maps look like in this region?

In the discussion the authors mention that the ZFs in Nos are unique compared to other RNA-binding ZFs. In what way?

Also in the discussion, the authors compare this cooperative interaction to that between Sxl and Unr. They say that this is also a cooperative interaction and two lines later they say it is additive. Which is it? If Sxl and Unr interaction is additive, then they are distinctly different.

These concepts make the whole story here, so it's very important that this is written clearly.

---

## [Author Response]

*The article by Weidman and colleagues addresses a long-lasting question regarding how the RNA binding proteins Pumilio and Nanos regulate their targets.*

*[…].*

*Though the structures and binding studies nicely illustrate how cooperativity works in this context, it would have been really worthwhile for authors to use this term early on and would not wait till the middle of their discussion.*

Thank you for this suggestion. We have now incorporated the term “cooperativity” into appropriate locations within the Abstract and Results sections.

*In addition, the paper would be improved if the interactions found*

*were discussed in terms of an evolutionary comparison. How conserved are these contacts? Can one use the structures to predict cooperativity of two RBPs in other organisms? This is not touched upon and would be very valuable.*

We have now addressed these interesting points in the Discussion:

“Based on our current understanding of the Pum-Nos interface and phylogenetic

comparisons, the Drosophila Nos-Pum interaction is conserved among Dipteran

orthologs, but it is not possible to predict conservation of the interaction in vertebrates.

[…]

Instead, the Nos and Pum contacts likely coevolved as the number of Nos and Pum homologs increased, perhaps restricting interactions between particular Nos and Pum proteins. Nos and Pum homologs may also interact indirectly, mediated by a bridging partner(s), as was suggested for C. elegans (Kraemer et al., 1999) and mouse (Suzuki et al. 2016).”

*Several issues should be clarified:*

*Much of the analysis is a comparison between the binary Pum-RNA complex and the ternary Nos-Pum-RNA complex. The structure of Pum changes in the region where Nos binds to it and to the RNA element. However, in the binary complex, the RNA doesn't really reach that area. It is shorter and contains only the PRE. Therefore, would Pum assume this altered conformation in the presence of a longer RNA without Nos? Or might it be closer to the conformation it assumes with Nos? This is an important point and bears on the conclusions of this paper.*

We thank the reviewers for their insightful questions about the structure of the C-terminal region of Pum in the binary complex. We conducted crystallization screens, but did not obtain crystals of Pum in complex with a *hb* RNA containing 4 additional upstream nucleotides. We therefore cannot rule out the possibility that the unfolding of the C-terminal α helix of Pum observed in the ternary complexes could be induced by a longer RNA as well as by Nos. Indeed the C-terminal region of Pum is involved in crystal packing in the binary complex that could stabilize the terminal α helix. We have edited the Results to clarify this point:

“Comparison of the ternary and binary complexes reveals that the addition of Nos and the upstream nucleotides induces localized conformational changes in Pum that promote Nos-Pum interaction and binding of Pum to RNA upstream of the core PRE site.”

And

“Since the upstream nucleotides were not present in the binary complex, the change in the structure of the C-terminal region of Pum may be induced by the presence of the upstream RNA and/or Nos. With these conformational changes, Pum and Nos interact with one another and together recognize RNA immediately 5´ of the core PRE motif.”

Although we feel it is not strong enough evidence to include in the manuscript, our previous structural studies of human Pum1, whose RNA-binding domain shares 80% sequence identity with *Drosophila* Pum, provide some indication that the C-terminal α helix remains formed in binary complexes with longer RNAs. We determined crystal structures of human Pum1 in complex with *hb* RNAs bearing five additional 5´ nucleotides. In one of the two Pum1:RNA complexes in the asymmetric unit, crystal packing influences the ordering of one additional upstream nucleotide. In the second complex, the upstream nucleotides were disordered and the C-terminal helix was formed as in the binary complex of Pum. The crystal packing in the area does not appear to restrict unfolding of the terminal helix or binding of the upstream RNA in the binary complex. The sequence of the C-terminal region that undergoes rearrangement bears only one substitution in Pum1 (N1421 is a leucine in human Pum1), making our observations in human Pum1 a reasonable model for *Drosophila* Pum.

*It is very hard to see from the various figures, what is moving in Pum between the binary and ternary complexes. Perhaps a side by side of that area would be easier to follow.*

Thank you for this feedback on the figures illustrating the differences in Pum between the binary and ternary complexes. We now include revised Figure 2—figure supplement 2 where we show side-by-side views of the two complexes. We changed the coloring to clarify the positions of changes we discuss, and present Nanos as a surface representation to focus on the R7/R8 loop and C-terminus of Pum. We also include an additional view of the Pum-Nos-*hb* RNA ternary structure in Figure 2—figure supplement 3, which highlights some of the new contacts in the ternary complex. These revised figures make it easier to see the differences and strengthen the clarity of our findings.

*The deletions that are discussed on Results section, subsection “Nanos increases the binding affinity of Pumilio for hunchback RNA”, that affect repression involve cutting out a few residues sometimes in the middle of a helix and it's not clear how well the structure would retain its integrity. The extent of repression might correlate with what percentage of folded protein they have in this region. Does this correlate with Pum-RNA binding?*

We chose the specific deletions because the Nos Δ376-382 is important in terms of history, biology, and function. This deletion corresponds to the lesion in the *nos^L7^* allele, which disrupts embryonic development. Together, these data and our functional and structural data reveal how that deletion causes loss of function. Unfortunately we were unable to produce recombinant purified Nos Δ376-382 protein to confirm this result but, as noted in the Discussion, Sonoda and Wharton reported the Nos Δ376-382 was unable to form a ternary complex with *hb* NRE, as assessed by yeast three hybrid assay. We have now clarified these points in the Results:

“Since deletion of the C-terminal region severely limited Nos repression of the hb 3´UTR reporter in cells (Figure 1), we probed this interaction more precisely. […] Since our crystal structure indicates that the deleted region includes part of the Cterminal helix that interacts with Pum, it is possible that the protein, although expressed, could be incorrectly folded.”

We have now performed additional analyses of the role of the observed Nos-Pum contacts by introducing single amino acid substitutions into the C-terminal region of Nos. As reported in the new Figure 4, Nos mediated repression was reduced by I376A and M378A substitutions, consistent with the contacts of these residues with Pum observed in the crystal structure. These results are now described on Line 195 of the Results section:

“We further examined this region by introducing single amino acid substitutions,

including the I376 and M378 residues that contact Pum in the structure (Figure 4). Repression activity of Nos I376A was diminished to 46%, and more so for Nos M378A (29%), whereas Nos I382A caused a modest decrease to 60%, relative to 70% repression by wild type Nos (Figure 4).”

We also tested mutants in the Nos binding interface of Pum to confirm the importance of the observed Nos-Pum interaction for ternary complex formation. These substitutions blocked incorporation of Nos into the complex, but did not prevent RNA binding by Pum, demonstrating that they did not cause an overall disruption of Pum structure.:

“We also found that single amino acid substitutions in Pum disrupt formation of the repression complex. […]Thus, Nos must interact with both repeat R7 and the R7-R8 loop of Pum to form a stable ternary complex.”

*The authors show a difference in the RNA structure between the hb and CycB complexes. However, the resolution is fairly low. How confident are they? What do the electron density maps look like in this region?*

We now include comparisons of the electron density maps for the *hb* and *CycB* ternary complexes, provided as Figure 6—figure supplement 5, which is cited in the Results. The electron density for the RNA in the *hb* and *CycB* complexes clearly indicates differences in the backbone conformations. The backbone conformation for the *CycB* RNA appears to change to accommodate the +6U nucleotide that interacts non-canonically with repeat R3 that bears an A-recognition motif. The map-model swaps demonstrate the differences in the two models.

*In the discussion the authors mention that the ZFs in Nos are unique compared to other RNA-binding ZFs. In what way?*

We thank the reviewers for pointing out that we should describe what we did to evaluate the Nos ZFs relative to other RNA-binding ZFs. We elaborate now in the text that no structural homologues were detected using the program DALI (other than the zebrafish Nos). We also considered a manual alignment with the HIV nucleocapsid protein Zn knuckles (ZK), which are also CCHC RNA-binding ZFs. We were pleased to discover that the topology and structures of Nos ZF2 and the HIV ZKs were strikingly similar. We have added a figure (Figure 9) to show this comparison. The following text was added to the manuscript.

*Also in the discussion, the authors compare this cooperative interaction to that between Sxl and Unr. They say that this is also a cooperative interaction and two lines later they say it is additive. Which is it? If Sxl and Unr interaction is additive, then they are distinctly different.*

These concepts make the whole story here, so it's very important that this is written clearly.

We thank the reviewers for pointing out that we should describe what we did to evaluate the Nos ZFs relative to other RNA-binding ZFs. We elaborate now in the text that no structural homologues were detected using the program DALI (other than the zebrafish Nos). We also considered a manual alignment with the HIV nucleocapsid protein Zn knuckles (ZK), which are also CCHC RNA-binding ZFs. We were pleased to discover that the topology and structures of Nos ZF2 and the HIV ZKs were strikingly similar. We have added a figure (Figure 9) to show this comparison. The following text was added to the Discussion:

“Nos uses its three functional regions: N, Z, and C (Figure 1) to elicit repression of target mRNAs, each of which illustrate principles of combinatorial control. […]However, we manually compared the Nos ZFs with CCHC Zn knuckles (ZKs) from HIV nucleocapsid protein (HIVnc) (De Guzman et al., 1998) and found that Nos ZF2 is strikingly similar to the HIVnc ZKs (Figure 9).”

Also in the discussion, the authors compare this cooperative interaction to that between Sxl and Unr. They say that this is also a cooperative interaction and two lines later they say it is additive. Which is it? If Sxl and Unr interaction is additive, then they are distinctly different.

*These concepts make the whole story here, so it's very important that this is written clearly.*

We are sorry for the confusion our abbreviated comparison caused. To clarify, we have now added a paragraph to the Discussion that compares and contrasts cooperative RNA recognition by Sxl and Unr with Nos and Pum, which is quite interesting and instructive of mechanisms of cooperativity:

“Cooperative RNA recognition by Nos and Pum is reminiscent of cooperative binding of msl2 RNA by Sex lethal (Sxl) and Upstream of N-Ras (Unr) (Hennig et al., 2014), yet the mechanisms and effects of cooperative recognition by Nos and Pum display novel distinguishing characteristics. […] As a consequence,

Nos and Pum cooperatively bind to RNAs that neither protein stably binds on its own.”